# MaskSearch: Towards Scalable Agentic Pre-Training for Search-Enhanced Reasoning

## Abstract

Retrieval-Augmented Large Language Models (LLMs) excel at knowledge-intensive tasks but struggle in complex scenarios due to passive retrieval. While search agents empower LLMs to actively use tools for reasoning, existing training-based methods remain constrained by task-specific data. Therefore, we propose **MaskSearch**, a two-stage training framework that bridges foundation models with search agents through a novel Retrieval-Augmented Mask Prediction (RAMP) task. Models first learn to recover masked spans via multi-step search and reasoning as a pre-training stage to endow foundational agentic capabilities that can be further improved by post-training on downstream tasks. We apply Supervised Fine-tuning (SFT) or Reinforcement Learning (RL) for training. For SFT, we combine multi-agent trajectory synthesis with iterative self-evolution distillation to construct data. For RL, we employ DAPO with a hybrid reward system consisting of answer and format rewards. Additionally, we introduce a curriculum learning strategy based on the number of masked spans. We evaluate the effectiveness of our framework in the scenario of open-domain multi-hop question answering. Extensive experiments demonstrate that MaskSearch effectively equips LLMs with transferable agentic abilities, advancing the development of search agents.

## 1 Introduction

Large Language Models (LLMs) (Yang et al., 2025; DeepSeek-AI et al., 2025; Dubey et al., 2024; Minaee et al., 2024) demonstrate strong performance across a variety of tasks leveraging its vast internal knowledge (Chang et al., 2024; Wu et al., 2023; Petroni et al., 2019), but suffer from hallucinations and often fall short in effectively addressing domain-specific or real-time tasks (Ding et al., 2024; Lewis et al., 2020). Retrieval-Augmented Language Models (RALMs) (Ram et al., 2023; Lewis et al., 2020; Guu et al., 2020) have been proposed to enhance LLMs by incorporating external knowledge, where a retrieval mechanism fetches information relevant to the input to augment the model's generation. While this approach has been successful, the separation of retrieval and generation limits the model's adaptability, preventing it from proactively acquiring information needed for multi-step tasks.

With autonomous AI agents gaining momentum, a new paradigm has emerged—web agents leveraging search engines as tools while employing agentic strategies such as planning, reasoning, reflection, and multi-agent collaboration to enhance their problem-solving capabilities. While prompt-based workflows have been widely used (Li et al., 2025b), they suffer from inefficiency and a lack of flexibility. Training an agentic model offers a more promising solution (Jin et al., 2025). However, current training methods are constrained by the limited size of requested task-specific datasets, which provide insufficient training signals. This limitation predisposes the model to narrow patterns and constrains its ultimate performance, highlighting the need for a scalable training paradigm.

Previous research on Masked Language Models (MLM) (Guu et al., 2020; Devlin et al., 2019) has proved that model capabilities can be incentivized through nearly infinite, diverse and verifiable data. Building on this insight, we propose **MaskSearch**, which enhances agentic search capabilities within a unified model architecture by taking a scalable Retrieval-Augmented Mask Prediction (RAMP) task as a **pre-training** stage. As illustrated in Figure 1, the model learns to fill masked spans by performing multi-step search instead of relying solely on the given context, aligning naturally with search agent mechanisms. This fosters foundational agentic capabilities including planning, tool

Figure 1: We propose a two-stage training paradigm for LLM-based search agents. Pre-training is conducted on a scalable and unlabeled Retrieval-Augmented Mask Prediction (RAMP) task, while post-training is performed on downstream open-domain multi-hop question answering tasks.

utilization and observation-based reasoning, which are highly transferable and can be consolidated during **post-training** on downstream tasks such as open-domain question answering.

Both Supervised Fine-tuning (SFT) and Reinforcement Learning (RL) can be applied in both stages of training. For SFT, it requires Chain-of-Thought (CoT) trajectories that explicitly outline the step-by-step process. We combine agent-based and distillation-based approaches to construct training data. Initially, a multi-agent system involving a planner, rewriter, and observer agent is utilized to synthesize reasoning data. Subsequently, we adopt an iterative self-evolution distillation strategy, using an increasingly fine-tuned model to generate the next dataset partition. For RL, we employ the Dynamic Sampling Policy Optimization (DAPO) algorithm (Yu et al., 2025) to optimize the model's search and reasoning process with a hybrid reward system including rule-based format rewards and model-based answer rewards. In addition, we introduce Curriculum Learning for RAMP based on the number of masks to gradually increase task difficulty during training.

Extensive experiments demonstrate that incorporating the RAMP-based **pre-training** stage significantly improves performance on both in-domain and out-of-domain multi-hop QA benchmarks. We further validate the scalability of MASKSEARCH on a 10M-scale RAMP dataset for SFT. These findings highlight the effectiveness of MASKSEARCH in enhancing agentic search capabilities of LLMs, regardless of their size, type, or the specific training method employed.

## 2 MASKSEARCH

### 2.1 PRELIMINARY

We consider an **LLM-based search agent** $\pi_\theta$ as a model capable of multi-step interaction with a search tool $\mathcal{R}$ to reason over proactively retrieved external information $D_\mathcal{R}$. To train $\pi_\theta$, which is parameterized by $\theta$ and initialized from a pre-trained base model to take a strong starting point, there are two primary methods: Supervised Fine-tuning (SFT) and Reinforcement Learning (RL).

**Supervised Fine-tuning (SFT)** Let $\pi_t$ denote an advanced implementation of agentic search. We construct a dataset $\mathcal{D}_t$ consisting of Chain-of-Thought (CoT) trajectories $y_t = \pi_t(x)$, which capture the reasoning and search steps to solve tasks. SFT trains $\pi_\theta$ to imitate $\pi_t$ by minimizing the discrepancy between the model output and the CoT trajectories:

$$\pi_\theta \leftarrow \arg\min_\pi \mathbb{E}_{(x,y_t)\sim\mathbb{D}_t} \left[\mathcal{L}(\pi_\theta(x, D_\mathcal{R}), y_t)\right] \tag{1}$$

During training, we treat the search results $D_\mathcal{R}$ as a latent variable by masking the retrieved tokens and focusing on optimizing the model outputs.

**Reinforcement Learning (RL)** We follow the paradigm of Search-R1 (Jin et al., 2025), which incorporates the search engine $\mathcal{R}$ into the RL process for optimization. The optimization objective is formulated as:

$$\max_{\pi_\theta} \mathbb{E}_{x\sim\mathcal{D}, y\sim\pi_\theta(\cdot|x;\mathcal{R})} \left[r_\phi(x, y)\right] - \beta\mathbb{D}_{\text{KL}} \left[\pi_\theta(y \mid x; \mathcal{R}) \,||\, \pi_{\text{ref}}(y \mid x; \mathcal{R})\right], \tag{2}$$

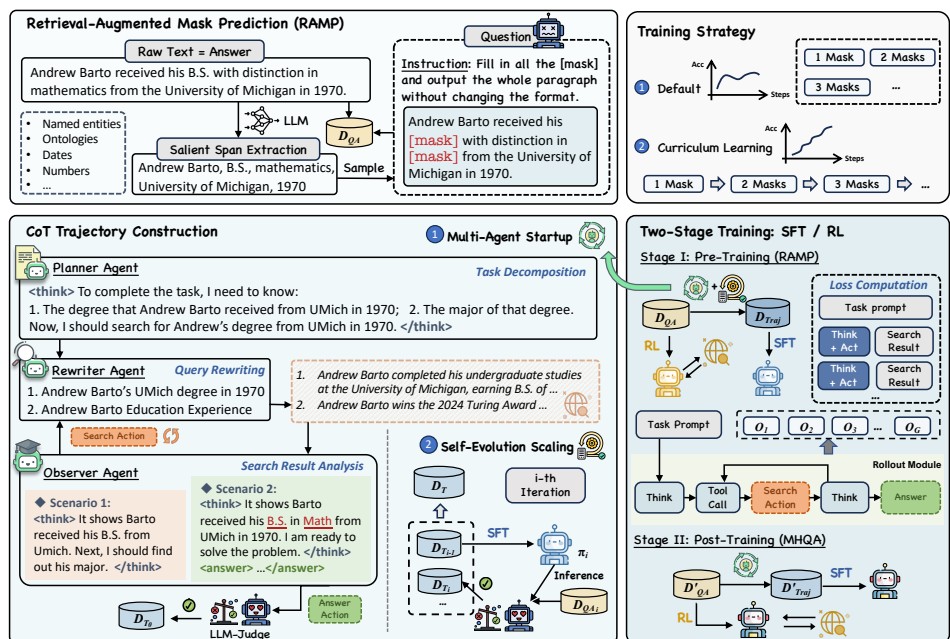

Figure 2: Overview of MASKSEARCH, where the model is first pre-trained on the RAMP task to establish foundational agentic search cabilities, enabling it to generalize effectively before downstream task training. Models can be trained via SFT or RL.

where $\pi_\theta$ is the policy model, $\pi_{\text{ref}}$ is the reference model, $r_\phi$ is the reward function and $\mathbb{D}_{\text{KL}}$ is KL-divergence measure. We employ the Dynamic Sampling Policy Optimization (DAPO) RL algorithms (Yu et al., 2025), and mask the retrieved tokens from the search engine during gradient computation as well. Details can be found in Appendix G.

## 2.2 RETRIEVAL-AUGMENTED MASK PREDICTION

We define the pre-training task of MASKSEARCH as *Retrieval-Augmented Mask Prediction* (RAMP). Given an input context sequence $x$, the objective is to predict a set of masked spans by proactively retrieving relevant information with a search tool $\mathcal{R}$ and reasoning over the retrieved results. This setting requires the model to engage in multi-step retrieval and reasoning, which bridges the gap between the base model and the search agent.

To construct questions and answers for RAMP, we leverage Wikipedia as the data source to ensure broad domain coverage and factual reliability. Since the answers naturally originate from the original paragraphs, our task is to design a method for selectively masking spans in the paragraphs to generate challenging questions, with a focus on spans that are not predictable from local context alone. **Salient Span Masking** (Cole et al., 2023; Guu et al., 2020) is employed to create challenging prediction targets, which correspond to knowledge-intensive elements that demand retrieval for reconstruction. We broaden the definition of salient spans beyond named entities and temporal expressions to also include ontologies, specific terms, and numerical values. Qwen-Turbo (Yang et al., 2025) is utilized to extract salient spans. The instruction is presented in Table 7.

From the extracted candidates, we randomly select $k$ spans (where $0 < k < 5$) and replace them with the special token [MASK]. This strategy balances task difficulty and tractability, ensuring that Chain-of-Thought (CoT) trajectories remain focused yet sufficiently challenging. The model is then trained to recover the original spans via multi-step interaction with $\mathcal{R}$.

## 2.3 COT TRAJECTORY CONSTRUCTION

Supervised Fine-tuning (SFT) on constructed CoT trajectories is the most direct way to enable multi-step reasoning. This involves generating reasoning traces that guide the student model through.

There are typically two ways to synthesize data: (1) agent-based, prompting a model to complete the task, and (2) distillation-based, using a stronger model to generate traces directly. We propose a hybrid approach for constructing CoT data that combines the two approaches, and construct a 10M CoT dataset (14B tokens) to validate the scalability of MASKSEARCH as a pre-training framework.

**Agent-Based Startup**   Initially, we orchestrate a synthesizing process involving planning, search, and reflection, supported by a multi-agent system, as shown in Figure 1. The **Planner Agent** first analyzes the overall task and breaks it into sub-tasks, generating an initial search query. The **Rewriter Agent** refines the generated query for improved knowledge retrieval and calls the search tool. The **Observer Agent** reviews the search results and steps taken, determining whether the task can be resolved or if additional steps are needed, updating the process until the final answer is obtained. Each agent is instructed with a few-shot prompt, as detailed in Appendix F.2. The final answer is evaluated by LLM-as-Judge and only trajectories that correctly fill all the masks are curated in $\mathbb{D}_0$.

**Self-Evolution Distillation**   To rapidly scale up the dataset while maintaining high data quality, we utilize an iterative generation strategy by using a trained teacher model $\pi_t$ instead of the multi-agent method. During the $j$-th iteration, the current dataset $\mathbb{D}_j$ is used to fine-tune the model $\pi_{\theta_{j-1}}$, resulting in an updated policy $\pi_{\theta_j}$. This updated policy incorporates the reasoning traces and search strategies learned from $\mathbb{D}_j$. The updated policy $\pi_{\theta_j}$ is then employed as the new teacher model $\pi_{t_{j+1}}$, which is used to synthesize the next iteration of the dataset $\mathbb{D}_{j+1}$.

$$\pi_{\theta_j} \leftarrow \arg\min_\pi \mathbb{E}_{(x,y_t)\sim\mathbb{D}_j} \left[ \mathcal{L}(\pi_{\theta_j}(x,D), y_t) \right], \quad \pi_{t_{j+1}} \leftarrow \pi_{\theta_j} \tag{3}$$

$$\mathbb{D}_{j+1} \leftarrow \{(x, y_t) \mid y_t = \pi_{t_{j+1}}(x, D)\}, \quad \mathbb{D}_t \leftarrow \{\mathbb{D}_0, \ldots, \mathbb{D}_{j+1}\} \tag{4}$$

This iterative approach ensures that the model continuously learns from increasingly complex and diverse reasoning traces, as each iteration builds on the improved capabilities of the previous one.

## 2.4   RL REWARD DESIGN

Reward signals are essential in RL, guiding the direction of model optimization. Our hybrid reward system consists of two components: a format reward and an answer reward, each contributing 50% to a total score of 1:

$$r_\phi(x, y) = 0.5R_f(y) + 0.5R_a(\hat{y}, y_{pre}) \tag{5}$$

where $\hat{y}$ is the ground truth answer, $y$ is the response and $y_{pre}$ is the extracted final answer from $y$. The format reward $R_f$ evaluates the response of the policy model to ensure it conforms to the specified answer format. It uses string matching to check whether the response contains special symbols that distinguish the various inference stages. If the response meets the format criteria, it is awarded 1 point; otherwise, it receives 0.

For the answer reward $R_a$, we explore various reward functions to identify the optimal one for optimization:

1) **Token-level Recall Reward**: We set token-level Recall $TR$ as the reward metric.

2) **Token-level Recall with Answer Length Penalty Reward** (Penalty-based Reward):   Recall Reward often lead to reward hacking. To address this, we introduce the penalty for the answer length, defined by the following:

$$R_a(\hat{y}, y_{pre}) = TR(\hat{y}, y_{pre}) - \alpha(\min(\max(\log_2\left(\frac{|y_{pre}|}{\beta \times |\hat{y}|}\right), 0), \gamma)) \tag{6}$$

where $\alpha$, $\beta$, $\gamma$ are parameters that adjust the strictness of the length penalty, set to 0.2, 8, and 4.

3) **Model-Based Reward**: We utilize the Qwen2.5-72B-Instruct model as a judge, evaluating the consistency between generated answers and standard answers, assigning a score of 0 or 1.

## 2.5   CURRICULUM LEARNING

Curriculum Learning (CL) is a training strategy that sorts training samples by difficulty and presents them to the model in an incremental manner. In the context of RAMP, the number of masked spans

$k$ serves as the primary metric for difficulty. Instead of random sampling, the curriculum learning method starts with simpler tasks containing fewer masked spans and progressively introduces more complex tasks with a higher number of masked spans. This approach allows the model to first learn fundamental reasoning skills and gradually build up its capabilities to handle more challenging scenarios. By following this curriculum, the model can better adapt to the increasing complexity of the tasks, leading to improved performance and more robust reasoning abilities.

$$\pi_\theta \leftarrow \arg\min_\pi \mathbb{E}_{(x,y_t)\sim\mathbb{D}_{t_k}} \left[ \mathcal{L}(\pi_\theta(x, D), y_t) \right], \quad \text{for } k \text{ in } \{1, 2, 3, 4\} \tag{7}$$

## 3 EXPERIMENT SETUP

**Models** We experiment with two model families: (1) **QWEN2.5** (Yang et al., 2025): QWEN2.5-1.5B, QWEN2.5-3B and QWEN2.5-7B; (2) **LLAMA3** (Dubey et al., 2024): LLAMA-3.2-1B, LLaMa-3.2-3B and LLAMA-3.1-8B. The multi-agent system is built upon QWEN-MAX to construct CoT trajectories. For self-evolution distillation, we fine-tune QWEN2.5-7B when the curated dataset size reaches 250K, 500K and 1M. Instruct models are used for RL, as base models poorly follow instructions. Detailed training setups are in Appendix B.

| Dataset | Hops | #Test |
|---|---|---|
| HotpotQA†(Yang et al., 2018) | 2 | 7405 |
| FanoutQA†(Zhu et al., 2024) | avg. 7 | 310 |
| Musique†(Trivedi et al., 2022) | 2-4 | 2417 |
| 2WikiMultiHopQA †(Ho et al., 2020) | 2-4 | 12576 |
| Bamboogle (Press et al., 2023) | 2 | 125 |
| FreshQA (Vu et al., 2023) | 1-2 | 374 |

Table 1: Test datasets used in the experiment. Datasets marked with † use their development split as the test set.

**Datasets and Metric** Beyond our RAMP pre-training data, we adopt open-domain multi-hop QA datasets for post-training and evaluation, which are widely recognized for testing search-enhanced reasoning capabilities. For **post-training**, we select HotpotQA (Yang et al., 2018) as a representative challenge and construct 58K correct CoT trajectories from its training set via the agent-based method for SFT. As summarized in Table 1, HotpotQA also serves as the in-domain test set. For out-of-domain testing, we evaluate on a variety of benchmarks, including FanoutQA (Zhu et al., 2024), Musique (Trivedi et al., 2022), 2WikiMultiHopQA (Ho et al., 2020), Bamboogle (Press et al., 2023), as well as FreshQA (Vu et al., 2023) after removing false-premise questions for single-hop reasoning. We report token-level Recall as the evaluation metric, which measures the overlap ratio between generated responses and ground-truth answers after segmenting both into token sequences.

**Baselines** We compare our method against the following baselines with retrieval: (1) **RAG-PE**: The model generates a response based on retrieval results; (2) **Agent-PE**: As elaborated in Section 2.3, it is a prompt-based multi-agent to handle complex reasoning tasks; (3) **Distilled Search-R1**: The model is directly fine-tuned on the downstream data, i.e., 58K CoT trajectories from HotpotQA; (4) **Search R1** (Jin et al., 2025): The model is directly trained using reinforcement learning on the downstream data from HotpotQA until its performance on the validation set converges.

## 4 EXPERIMENTAL RESULTS

### 4.1 MAIN RESULTS

**First-stage training of the RAMP task benefits LLM-based search agents.** As shown in Table 2, our proposed MASKSEARCH significantly enhances the model performance across multiple open-domain question-answering datasets. In the in-domain dataset, i.e., HotpotQA, RAMP provides a stable improvement in the model's recall score. On out-of-domain datasets, the improvement is even more pronounced. For instance, on the Bamboogle dataset, the Qwen2.15-1.5B model achieves a substantial increase of **11.78** compared to post-training only, while the LLaMA model sees an impressive gain of **15.12** [1]. Moreover, smaller models (e.g., Qwen2.5-1.5B) can perform comparably to larger models on a variety of tasks. This proves that RAMP, as a scalable learning signal, can help models strengthen their abilities to decompose problems and interact with search tools.

**RL offers higher performance gains compared with SFT, especially for larger models.** While SFT proves to be effective in improving agentic search capabilities, RL demonstrates the potential

---

[1]LLaMA results are presented in Appendix E.

| Methods | Pre-Train | Post-Train | HotpotQA | FanoutQA | Musique | 2Wiki | Bamboogle | FreshQA | Avg. |
|---|---|---|---|---|---|---|---|---|---|
| | | | | *Qwen2.5-1.5B* | | | | | |
| RAG-PE | ✗ | ✗ | 29.45 | 27.37 | 12.07 | 37.32 | 22.93 | 41.19 | 28.37 |
| Agent-PE | ✗ | ✗ | 48.74 | 36.15 | 28.87 | 49.72 | 58.87 | 63.27 | 47.60 |
| Distilled Search-R1 | ✗ | SFT | 64.13 | 47.76 | 35.02 | 76.45 | 63.87 | 68.12 | 59.22 |
| Search-R1 | ✗ | RL | 61.72 | 42.68 | 35.63 | 64.40 | 64.59 | 74.44 | 57.24 |
| | SFT | ✗ | 58.64 | 49.05 | 31.93 | 70.90 | 53.00 | 66.42 | 54.99 |
| | RL | ✗ | 58.91 | 40.12 | 27.67 | 65.97 | 47.93 | 73.44 | 52.34 |
| MASKSEARCH | SFT | SFT | 67.58 | **53.18** | 38.58 | **81.13** | **75.65** | 75.47 | 65.27 |
| | RL | SFT | 66.23 | 45.61 | 40.55 | 71.77 | 73.79 | 75.91 | 62.31 |
| | SFT | RL | 65.95 | 48.64 | 40.36 | 72.00 | 72.67 | 73.54 | 62.19 |
| | RL | RL | **71.02** | 49.73 | **40.59** | 76.03 | 73.60 | **81.08** | **65.34** |
| | | | | *Qwen2.5-3B* | | | | | |
| RAG-PE | ✗ | ✗ | 38.37 | 41.48 | 20.78 | 51.14 | 37.60 | 61.55 | 41.82 |
| Agent-PE | ✗ | ✗ | 51.17 | 49.82 | 25.27 | 58.14 | 56.40 | 67.80 | 48.10 |
| Distilled Search-R1 | ✗ | SFT | 67.38 | 54.00 | 38.20 | 79.76 | 68.05 | 77.59 | 64.17 |
| Search-R1 | ✗ | RL | 69.03 | 48.55 | 39.08 | 78.85 | 72.53 | 76.78 | 64.14 |
| | SFT | ✗ | 62.45 | 51.76 | 35.51 | 75.88 | 60.00 | 74.26 | 59.98 |
| | RL | ✗ | 64.96 | 53.53 | 34.11 | 67.87 | 73.33 | 75.32 | 61.52 |
| MASKSEARCH | SFT | SFT | 69.30 | **56.03** | 40.12 | 82.36 | 74.52 | 79.84 | 67.03 |
| | RL | SFT | 68.23 | 55.85 | 39.48 | 81.72 | 73.87 | 77.58 | 66.12 |
| | SFT | RL | 69.03 | 54.55 | 39.08 | 79.85 | 72.53 | 76.78 | 65.30 |
| | RL | RL | **73.08** | 53.02 | **44.48** | 80.43 | **80.13** | **85.07** | **69.37** |
| | | | | *Qwen2.5-7B* | | | | | |
| RAG-PE | ✗ | ✗ | 43.55 | 51.92 | 25.05 | 53.86 | 44.60 | 64.40 | 47.23 |
| Agent-PE | ✗ | ✗ | 61.75 | 55.69 | 34.25 | 68.77 | 63.25 | 75.81 | 58.25 |
| Distilled Search-R1 | ✗ | SFT | 69.55 | 57.24 | 41.06 | 83.84 | 73.07 | 78.97 | 67.29 |
| Search-R1 | ✗ | RL | 70.59 | 56.25 | 41.29 | 80.50 | 79.33 | 78.46 | 67.74 |
| | SFT | ✗ | 63.84 | 56.22 | 38.06 | 75.36 | 62.15 | 74.75 | 61.73 |
| | RL | ✗ | 67.77 | 54.68 | 35.98 | 75.61 | 70.27 | 76.37 | 63.44 |
| MASKSEARCH | SFT | SFT | 70.44 | **60.85** | 41.76 | **84.65** | 80.13 | **81.12** | 69.83 |
| | RL | SFT | 70.84 | 56.29 | 41.90 | 83.38 | 78.53 | 78.93 | 68.31 |
| | SFT | RL | 71.69 | 57.69 | 42.23 | 81.25 | 81.87 | 75.42 | 68.36 |
| | RL | RL | **75.61** | 58.96 | **45.54** | 82.10 | **83.00** | 80.85 | **71.01** |

Table 2: Evaluation results of different methods on various open-domain question answering datasets. Bold and underlined indicate the best and the second-best results.

to achieve even higher upper limits of MASKSEARCH. During training on RAMP, both 3B and 7B models show that RL can surpass SFT, highlighting its advantage in optimizing retrieval-augmented reasoning. Notably, the gains from applying RL in both stages of training are more pronounced on the 7B model compared to the 1.5B model, aligning with prior findings that RL better unlocks the potential of larger models.

**Consistency across different stages improves performance.** We observe that using the same training method in both stages—whether SFT→SFT or RL→RL—consistently outperforms mixed pipelines such as SFT→RL. We assume this is because maintaining a consistent objective across pre-training and post-training ensures stable optimization and coherent credit assignment. In contrast, switching methods between stages introduces conflicts between objectives, which can hinder exploration and require substantial policy adjustment, limiting overall gains.

## 4.2 SCALING PERFORMANCE

To assess MASKSEARCH's scaling potential, we vary pre-training steps and subsequently fine-tune models on downstream tasks. As illustrated in Figure 3, small models exhibit substantial gains from the additional pre-training and mark an explicit scaling law. For larger models, scaling up the dataset size remains beneficial, but the improvements are less pronounced than those observed for smaller models. The performance curve of the 7B model indirectly reflects the limitations of self-evolution training data: while pre-training still yields gains over no pre-training, the benefits are bounded by the limited diversity and complexity of the model's own generations. This observation indicates that, although data quantity drives part of the performance increase, the quality and heterogeneity of

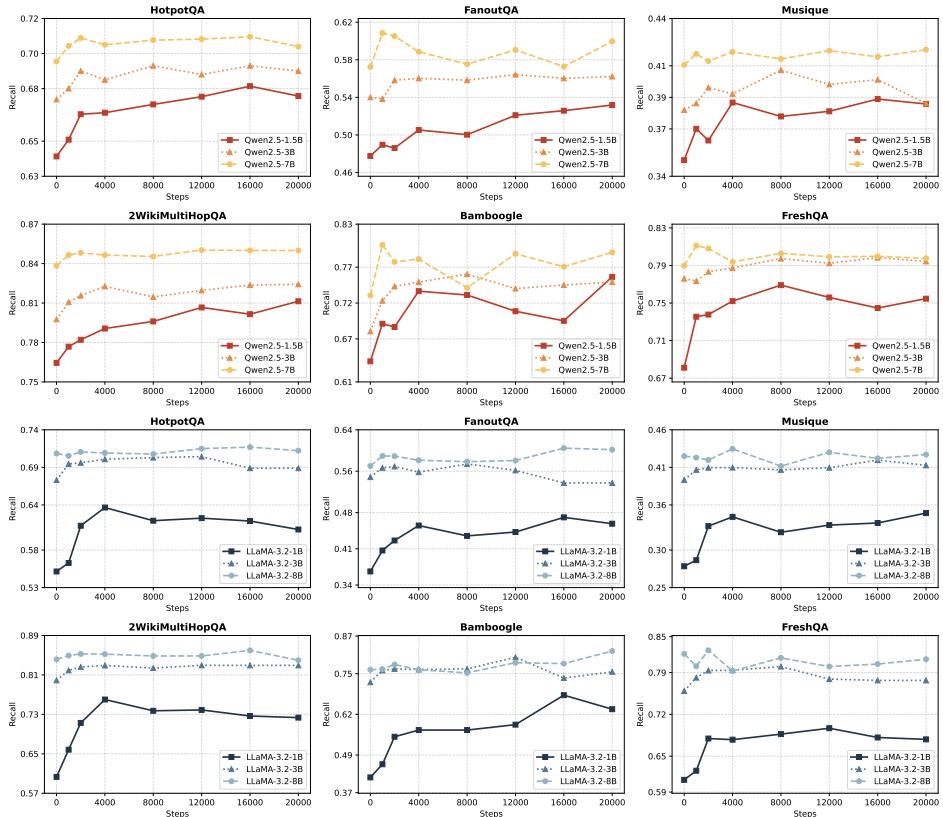

Figure 3: Scaling Performance of SFT with respect to training steps on RAMP. The results obtained at training step 0 align precisely with those of the *Direct SFT* baseline.

training data are critical determinants of the ultimate effectiveness of SFT. Overall, MASKSEARCH provides an effective and scalable pre-training objective for agentic LLMs.

### 4.3 SUPERVISED CURRICULUM LEARNING

We delve deeper into the effectiveness of curriculum learning (CL) in RAMP. We sample 10K data from RAMP for each number of masked spans and 6K from HotpotQA to maintain an appropriate ratio between pre-training and downstream tasks. Additionally, 500 QA pairs are sampled from the remaining RAMP data as a validation set, with 100 data points for each number of masked spans. We find that increasing the number of masked spans consistently improves validation performance, as depicted in Figure 4. Although the initial performance lags behind, CL ultimately outperforms the default mixed training strategy, with gains transferring to downstream tasks. Table 3 shows that CL benefits both Qwen2.5-7B and LLaMA-3.2-1B, demonstrating its generality across different models.

## 5 DISCUSSION

In this section, we conduct an in-depth discussion of the critical factors of RAMP and its training process to offer a comprehensive exploration of our approach. A case study is presented in the Appendix H to provide further insights into the practical application of our method.

### 5.1 MASKING STRATEGY

We delve into the impact of masking methods on the RAMP task. Beyond salient span masking, prior work suggests that more challenging masks—measured by Pointwise Mutual Information (PMI) or Perplexity (PPL)—can benefit MLMs. For autoregressive models, we implement a PPL-based

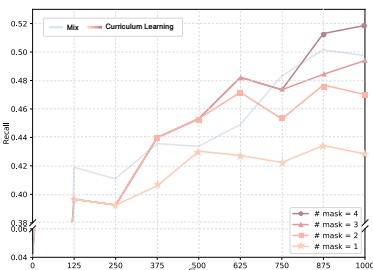

Figure 4: Performance on the dev set while finetuning with varying numbers of masks.

| Model | w/ RAMP | | w/o RAMP |
| --- | --- | --- | --- |
| | CL | Mix | |
| *Qwen* | | | |
| Qwen2.5-1.5B | 54.06 | **55.36** | 52.54 |
| Qwen2.5-7B | **65.42** | 65.36 | 64.35 |
| *LLaMA* | | | |
| LLaMA-3.2-1B | **55.93** | 53.67 | 52.98 |
| LLaMA-3.1-8B | 64.93 | **65.57** | 64.21 |

Table 3: Average recall of different training strategies on the test datasets.

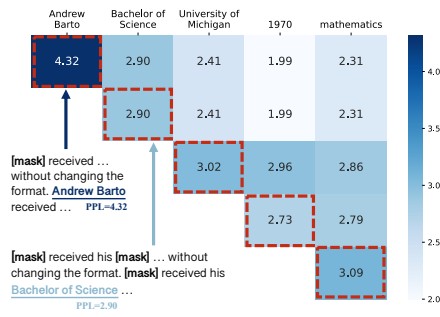

(a) PPL-based Masking Strategy.

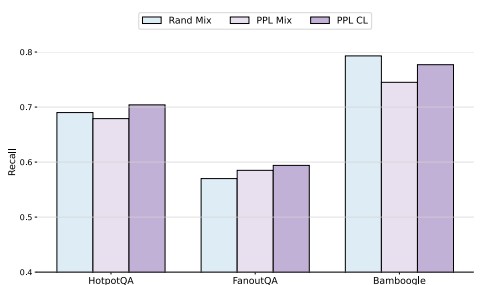

(b) Downstream Performance.

Figure 5: Exploring RAMP with PPL-based masking strategy. The subfigure (a) illustrates the computation of perplexity for the unmasked spans in each round. The subfigure (b) indicates the effect of the PPL-based masking strategy on downstream test sets along with CL.

masking strategy (Figure 5a), where spans with the highest perplexity are greedily selected until the masking budget is met. To validate this strategy against the original random masking strategy, we prepare a 40K RAMP dataset for SFT using our agent-based trajectory construction method and sample 6K HotPotQA trajectories for the second-stage training.

As shown in Figure 5(b), the PPL-based masking strategy yields a performance boost on the FanoutQA dataset, underscoring the efficacy of selecting more challenging masks. However, the experimental outcomes on the HotpotQA and Bamboogle datasets imply that merely augmenting the difficulty of the masked spans may result in a decline in performance. This suggests that the model may not have genuinely mastered the tasks due to its increased difficulty. On the other hand, curriculum learning, which progressively escalates the complexity of the training tasks, proves to be advantageous. After the second stage of training, the model demonstrates enhanced performance across all three datasets, highlighting its effectiveness in enhancing the model's learning progress to tackle intricate tasks.

## 5.2 PERFORMANCE WITH DIFFERENT RL REWARD

We study the impact of different RL rewards on model performance, as discussed in section 2.4. The results are shown in Figure 6, model trained with the token-level recall reward hacks the metric by adding a lot of irrelevant information to the answer, significantly increasing the length of the response. This results in a notable decline in actual performance compared to other RL rewards under model-based evaluation. While penalty-based reward can substantially reduce answer length, performance is still affected and the model still can hack within the constraint of limited answer length by employing enumeration in our observation. Model-based reward offer significant improvements of 34.71 and 19.48 over the other two reward methods, effectively addressing issues of reward hacking and demonstrating remarkable stability and effectiveness in RL training. These advantages ultimately led us to adopt the model-based reward approach for training.

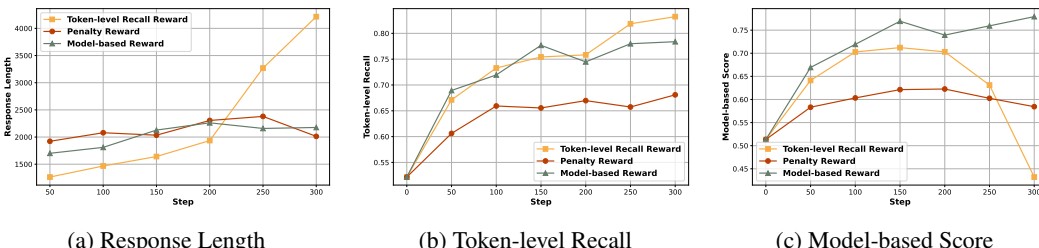

(a) Response Length       (b) Token-level Recall       (c) Model-based Score

Figure 6: Exploring Different Answer Reward Designs. These figures report the performance of Qwen2.5-7B on the development set during RL training: The subfigure (a) illustrates the length of responses generated by the model; (b) displays the token-level recall; (c) details the metric obtained from Qwen2.5-72B serving as an external judge.

## 6 RELATED WORK

**Retrieval-Augmented Generation** RAG bridges static model parameters with dynamic external knowledge (Ding et al., 2024; Yan et al., 2024; Gao et al., 2023; Lewis et al., 2020). Utilizing real-time or domain-specific (Zhang et al., 2024b; Gilbert et al., 2024) data, LLMs can mitigate issues like hallucinations (Li et al., 2024; Huang et al., 2023) and outdated facts. Early advances demonstrate that coupling neural retrievers with language models improves factual accuracy in tasks such as open-domain question answering (Lewis et al., 2020; Guu et al., 2020). Recent studies have advanced RAG by embedding autonomous AI agents into the RAG workflow (Singh et al., 2025; Ravuru et al., 2024; An et al., 2024), leveraging the advantage of agent frameworks like planning, reflection and multi-agent collaboration (Wang et al., 2025).

**Agent** LLM-based agents can autonomously reason and interact with dynamic environments, enabling applications from web search (He et al., 2025; Alzubi et al., 2025) and operations (Gur et al., 2023), creative writing (Zhang et al., 2024a), and travel planning (Gundawar et al., 2024). In addition to prompt engineering (Shinn et al., 2024; Park et al., 2023; Yao et al., 2023b; Jiabin Tang, 2025), recent research has also focused on optimizing and training these agent skills in an end-to-end manner (Putta et al., 2024; Feng et al., 2024). We build on these ideas to train models to enhance their agentic search and reasoning capabilities ==and introduce a training paradigm shift in agentic pre-training methodology.== The comparison with existing works is in Appendix D.

**Reasoning LLM** Reasoning is a central focus in LLM research for tackling complex tasks. Large Reasoning Models (LRMs) (Li et al., 2025a; DeepSeek-AI et al., 2025; Jaech et al., 2024) demonstrates the effectiveness of inference-time scaling (Muennighoff et al., 2025; Snell et al., 2024) and multi-step reasoning (Wei et al., 2022; Yao et al., 2023a;b) mechanisms. Beyond prompt-based methods (Li et al., 2025b; Fu et al., 2023b) and Supervised Fine-tuning (SFT) (Lin et al., 2024; Mukherjee et al., 2023), where even small models show competitive performance on multi-step reasoning (Srivastava et al., 2025; Shridhar et al., 2023; Fu et al., 2023a), Reinforcement Learning (RL) has emerged as a leading approach for advanced reasoning (Yu et al., 2025; Jin et al., 2025; DeepSeek-AI et al., 2025). In this work, we generate reasoning trajectories via prompt-based methods and combine and compare the effectiveness of SFT and RL during training.

## 7 CONCLUSION

In this paper, we introduce a novel training objective MASKSEARCH for enhancing the agentic search capabilities of LLMs. Our method, built around the Retrieval-Augmented Mask Prediction (RAMP) pre-training task, enables models to autonomously perform multi-step search and reasoning to fill in masked spans, fostering a deeper integration of external knowledge. We demonstrate that by using both Supervised Fine-tuning (SFT) and Reinforcement Learning (RL), along with curriculum learning, our framework leads to substantial performance improvements across various open-domain question answering tasks, compared to the baselines. Overall, our work underscores the potential of pre-training-style signals for foundation LLMs transitioning into agentic models with search-enhanced reasoning capabilities, providing a scalable framework of search agents across model sizes and architectures. The limitation of MASKSEARCH is discussed in Appendix K.

ETHICS STATEMENT

In this paper, we conduct experiments using publicly available datasets, including Hotpot, FanoutQA, Musique, 2WikiMultiHopQA, Bamboogle, and FreshQA, in accordance with their respective usage terms and conditions. We assure that no personally identifiable information or sensitive attributes are collected or re-distributed. The proposed method is a general-purpose reading-comprehension technique; we do not foresee direct misuse beyond the risks already present in existing QA systems. Nevertheless, large-scale generative models can surface inappropriate or biased text when deployed in the wild. We therefore recommend that downstream developers incorporate safety filters and bias-mitigation layers before deployment. No human subjects were involved in any stage of this research, and no additional IRB approval was required.

REPRODUCIBILITY STATEMENT

To facilitate full reproduction of our results, we provide the following resources in the supplementary ZIP: (1) scripts to generate the RAMP dataset from the raw Wikipedia corpus, including the Salient Span Extraction process; (2) the multi-agent system used to construct CoT trajectories; and (3) a preview of the RAMP dataset. The complete dataset and code will be publicly released in the future.

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

## A    USE OF LLMS

In adherence to ICLR 2026 policy, we report the assistive use of Large Language Models (LLMs) in this work. Specifically, ChatGPT was employed solely for grammatical corrections and language polishing, with all text carefully reviewed to ensure that no technical content, nuance, or citations were altered. Additionally, LLMs functioned as the search agents for performing RAMP and MHQA tasks. The authors have thoroughly examined all contributions generated with LLM assistance and take full responsibility for the integrity of this work.

## B    TRAINING SETTINGS

**Supervised Fine-tuning**    During the agentic pre-training phase of the scaling experiment, we ensure a global batch size of 1024. For models with 1B, 1.5B, and 3B parameters, we use a distributed training setup with 16 nodes and 8 H20 GPUs per node. For models with 7B and 8B parameters, we use a distributed training setup with 32 nodes and 8 GPUs per node. Each model is trained for 2 epochs with a learning rate set to $4 \times 10^{-5}$. For downstream task training, we use a global batch size of 64 with a learning rate set to $1 \times 10^{-5}$, training on 8 GPUs per node.

**Reinforcement Learning**    For DAPO training, we configure the batch size to 16, set the policy model learning rate to $1 \times 10^{-6}$, and sample 16 responses per prompt. The training for the CPT and downstream tasks is performed on 8 H20 GPUs over 1 epoch and early stops training when there is no better result on the development set for 150 steps. We use the instruct models for RL, as base models often fail to follow the instructions.

## C  COMPUTATIONAL COST ANALYSIS

We analyze the computational cost introduced by the proposed agentic pre-training stage, comparing it against (a) agentic post-training and (b) the original base-model pre-training.

### C.1  AGENTIC PRE-TRAINING VS. AGENTIC POST-TRAINING

To quantify the overhead of agentic pre-training, we report the wall-clock training time for both SFT and RL stages across different model scales. From Table 4, we observe that the cost of agentic pre-training (SFT + RL) is comparable to or moderately larger than that of agentic post-training, depending on model size. Importantly, unlike post-training, the agentic pre-training stage is *model-agnostic* and can be reused across multiple downstream tasks, effectively amortizing its one-time computational cost.

| Model Size | Pre-Train (SFT) | Pre-Train (RL) | Post-Train (SFT) | Post-Train (RL) |
|---|---|---|---|---|
| 7B | 4 days 14 hours | 5 days | 11 hours | 2 days |
| 3B | 2 days 7 hours | 3 days | 8 hours | 1 day 4 hours |
| 1.5B | 22 hours | < 2 days | 6 hours | 20 hours |

Table 4: Wall-clock training time for agentic pre-training and post-training across model scales.

### C.2  AGENTIC PRE-TRAINING VS. BASE-MODEL PRE-TRAINING

The Qwen pre-training corpus contains approximately 18T tokens Yang et al. (2025), while our RAMP agentic pre-training uses only up to 25B tokens—over two orders of magnitude smaller. Given this large gap in pre-training corpus size, the computational cost of RAMP agentic pre-training is negligible compared to full LLM pre-training, while still providing significant improvements in downstream agentic reasoning capabilities.

## D  DETAILED COMPARISON WITH EXISTING RALMs

MASKSEARCH introduces several innovations that distinguish it from prior work in the field of Retrieval-Augmented Language Models (RALMs) and search-enhanced reasoning models. Firstly, in terms of the number of retrieval tokens, MASKSEARCH operates with an effectively infinite retrieval space, which is a significant departure from traditional models that are limited by the size of their pre-defined retrieval corpora. Secondly, MASKSEARCH employs a single-model architecture, in contrast to the dual-model setups of traditional RALMs. This simplification not only reduces computational overhead but also enhances the coherence and consistency of the reasoning process. By interacting with a search engine in agentic style, MASKSEARCH supports multi-step reasoning, a capability that is not present in most of the models listed in the table. This feature is crucial for handling complex queries that require the integration of information from multiple sources and cannot be solved in a single step.

The work most similar to MASKSEARCH is Search-R1, but there are several key differences that set MASKSEARCH apart. While both models operate with an effectively infinite retrieval space and support multi-step reasoning, MASKSEARCH employs a two-stage end-to-end training process that can be adapted to both SFT and RL. This contrasts with Search-R1, which uses a single-stage reinforcement learning approach for its retriever. The two-stage end-to-end training in MASKSEARCH allows for more general optimization of the agent capabilities, leading to better integration with the LLM and improved performance.

## E  RESULTS FOR LLaMA MODELS

In this section, we present the results of our experiments conducted on varying sizes of LLaMA models. Our experiments were specifically carried out on the SFT, as is shown in Table 6. The main findings are summarized as follows:

| | # Retrieval Tokens | # Models | Retriever | Multi-Step |
|---|---|---|---|---|
| KNN-LM Khandelwal et al. (2020) | $O(10^9)$ | 2 | Transformer (Frozen) | ✗ |
| REALM Guu et al. (2020) | $O(10^9)$ | 2 | BERT (2-Stage E2E) | ✗ |
| RAG Lewis et al. (2020) | $O(10^9)$ | 2 | DPR (E2E) | ✗ |
| Retro Borgeaud et al. (2022) | $O(10^{12})$ | 2 | BERT (Frozen) | ✗ |
| Atlas Izacard et al. (2022) | $O(10^{10})$ | 2 | Contriever (E2E) | ✗ |
| IC-RALM Ram et al. (2023) | $O(10^8)$ | 2 | BM25 | ✗ |
| RADIT Lin et al. (2024) | $O(10^{11})$ | 2 | DRAGON+ (Separate) | ✗ |
| Search-R1 Jin et al. (2025) | ∞ | 1 | LLM (1-Stage RL) | ✓ |
| MASKSEARCH | ∞ | 1 | LLM (2-Stage E2E) | ✓ |

Table 5: Comparison with existing RALMs and search-enhanced reasoning models. E2E is short for end-to-end.

**Significant Performance Improvement over Baselines**   Similar to Qwen models, our experiments demonstrated that the MASKSEARCH framework achieved substantial performance improvements across multiple datasets compared to existing methods. On the LLaMA-3.2-1B model, MASKSEARCH achieved an average performance improvement of nearly 10 points over the next best method (Distilled Search-R1). This significant enhancement highlights the effectiveness of MASKSEARCH in improving search capabilities.

**Scalability Across Model Sizes**   The MASKSEARCHframework showed consistent performance improvements across different sizes of LLaMA models, ranging from 1B to 8B parameters. However, the extent of improvement for the SFT stage depends on the capability gaps between the distilled model and the student model, which is also observed with the Qwen series.

**Transferability among Different Model Series**   The MASKSEARCH framework showed strong transferability among different architectures of models, as we utilize data generated by the Qwen-series model but still achieve significant performance improvements for LLaMA models. This demonstrates the framework's ability to leverage data from different sources and adapt to different model architectures, further enhancing its applicability in diverse research and development scenarios.

Table 6: Complete results of LLaMA models.

| Methods | Pre-Training | Post-Training | HotpotQA | FanoutQA | Musique | 2Wiki | Bamboogle | FreshQA | Avg. |
|---|---|---|---|---|---|---|---|---|---|
| *LLaMA-3.2-1B* | | | | | | | | | |
| RAG-PE | ✗ | ✗ | 20.00 | 29.91 | 9.03 | 40.15 | 13.23 | 42.74 | 32.51 |
| Agent-PE | ✗ | ✗ | 37.51 | 31.73 | 19.14 | 45.96 | 32.22 | 57.24 | 37.30 |
| Distilled Search-R1 | ✗ | SFT | 55.15 | 36.62 | 27.84 | 60.32 | 41.87 | 61.05 | 47.14 |
| MASKSEARCH | SFT | SFT | 63.66 | 45.50 | 34.41 | 76.01 | 56.99 | 67.77 | 57.40 |
| *LLaMA-3.2-3B* | | | | | | | | | |
| RAG-PE | 34.00 | 48.84 | 15.44 | 53.79 | 46.35 | 63.33 | 48.63 | | |
| Agent-PE | 51.90 | 37.17 | 42.66 | 71.44 | 48.46 | 58.06 | 51.61 | | |
| Distilled Search-R1 | ✗ | SFT | 67.31 | 54.86 | 39.33 | 79.91 | 72.25 | 75.91 | 64.93 |
| MASKSEARCH | SFT | SFT | 70.31 | 57.40 | 40.67 | 82.39 | 76.52 | 80.00 | 67.88 |
| *LLaMA-3.1-8B* | | | | | | | | | |
| RAG-PE | 36.67 | 52.38 | 19.74 | 51.52 | 51.04 | 63.33 | 50.06 | | |
| Agent-PE | 59.70 | 52.30 | 41.12 | 77.09 | 59.69 | 71.85 | 60.29 | | |
| Distilled Search-R1 | ✗ | SFT | 70.85 | 57.01 | 42.50 | 84.15 | 76.27 | 82.13 | 68.82 |
| MASKSEARCH | SFT | SFT | 71.70 | 60.46 | 42.20 | 85.98 | 78.22 | 80.42 | 69.83 |

# F   PROMPT DEMONSTRATION

## F.1   SALIENT SPAN EXTRACTION FOR

Table 7: Prompt for salient mask extraction.

**Prompt:**

Extract all the knowledge entities (more than 5 entities if exist) from the following text. Knowledge entities refer to specific individuals, locations, organizations, conferences, journals, short academic concepts, and unique short numbers mentioned in the text. If multiple entities appear consecutively, only extract the finest-grained entity. Please output the extracted entities as a list directly, without any other content. Each entity should not exceed 10 characters.

If an entity repeatedly appears, you should not extract it. You need extract a whole word like [American] instead of [America]n.

Ensure that the entities you extract cannot be directly infer from the context of the paragraph. You must need extra information search on the Internet to infer the entities.

**Example 1:**

Paragraph: Recently, the team of Kou Xufen and Zhu Zhifeng from the Post-Moore Center (PMICC) at the School of Information Science and Technology, ShanghaiTech University, designed and prepared a 2-inch magnetic topological heterojunction Bi2Te3/CrTe2 thin film based on molecular beam epitaxy technology. They achieved a spin-orbit torque device array capable of possessing both brain-like synapse and neuron functions, and integrated batch normalization algorithms and trainable activation functions. The relevant research results were published online in the renowned academic journal ACS Nano under the title "Integrated Artificial Neural Network with Trainable Activation Function Enabled by Topological Insulator-based Spin-Orbit Torque Devices".

Entities: ["ShanghaiTech University", "Kou Xufen", "Zhu Zhifeng", "spin-orbit torque device array", "ACS Nano"]

**Example 2:**

Paragraph: For open addressing, the load factor is a particularly important factor and should be strictly limited below 0.7-0.8. Beyond 0.8, CPU cache misses (cache missing) increase exponentially when looking up tables. Therefore, some hash libraries that use open addressing, such as Java's hash libraries, limit the load factor to 0.75, and the hash table will be resized when this value is exceeded.

Entities: ["load factor", "exponential curve", "Java", "hash table", "0.75"]

**Example 3:**

Paragraph: Dalian buses are mainly 12 meters long, with the smallest currently being 6 meters. Pure electric vehicles are gradually taking the dominant position. The main model is the BYD K9 series pure electric bus, and Dalian public transportation has begun to purchase a large number of BYD pure electric buses. Diesel, natural gas, and hybrid vehicles are gradually exiting the stage of Dalian public transportation.

Entities: ["BYD K9", "6", "12"]

**Task:**

Paragraph: {paragraph}
Entities:

## F.2 REASONING TRAJECTORY CONSTRUCTION

This section presents the prompts used for agent-based reasoning trajectories construction, including the planner agent, rewriter agent, observer agent and LLM-Judge.

Table 8: Prompt for agent-based CoT trajectory construction.

**Planner Agent**

Your task is to provide the steps for solving a multi-hop search problem.
The output format should be: "`<think>` {Overall thought process} `</think>`
`<search>` [{"query": "query", "intent": 1}] `</search>` "; query should be in sentence format.

Here are some examples:

Table 8: Prompt for agent-based CoT trajectory construction.

**Question:** What is the undergraduate school of the director of the movie "Sense and Sensibility"?
**Thought:** <think> To answer this question, I will take the following steps:
1. First, find out who directed the movie "Sense and Sensibility".
2. Investigate the educational background of the director, particularly their undergraduate school.
3. Identify the specific institution where the director completed their undergraduate studies.
Now, I will start with the first step and search for the director of the movie "Sense and Sensibility". </think>
<search> [{"query": "Who is the director of the movie 'Sense and Sensibility'?", "intent": 1}] </search>

**Question:** When did the birthplace of the performer of Live and Beyond become the capital of the state where Knowles is located?
**Thought:** <think> To determine when the birthplace of the performer of Live and Beyond became the capital of the state where Knowles is located, I will take the following steps:
1. First, identify who the performer of Live and Beyond is.
2. Then find out the birthplace of this performer.
3. Next, search for which state Knowles is from.
4. Finally, determine when the birth city of the performer of Live and Beyond became the capital of Knowles' state.
Now, I will start with the first step and search online to determine who the performer of Live and Beyond is. </think>
<search> [{"query": "Find out who the performer of Live and Beyond is", "intent": 1}] </search>

**Question:** {input}
**Thought:**

---

**Rewriter Agent**

---

Given a piece of content containing queries to search, your task is to rewrite the queries in order to obtain more comprehensive search results. Please provide at least three rewritten queries. The output format should be the following JSON structure:
{"queries": ["query 1", "query 2", "query 3"]}

Here are some examples:

**Content:** <think> To find out which undergraduate school the director of the movie "Sense and Sensibility" attended, I will take the following steps:
1. First, determine who the director of the movie "Sense and Sensibility" is.
2. Then, search for educational background of this director, particularly undergraduate education.
Now, I'll proceed with the first step by using online searches to identify the director of the movie "Sense and Sensibility". </think>
<search> [{"query": "Find out who the director of the movie 'Sense and Sensibility' is", "intent": 1}] </search>
**Rewritten Queries:**
{"queries": ["Sense and Sensibility director", "Sense and Sensibility 1995 director", "Sense and Sensibility Filmmaker"]}

**Content:** <think> After analyzing the search results in detail, I concluded that the director of the movie "Sense and Sensibility" is Ang Lee. Therefore, I will proceed with the next step, where I need to search for his detailed undergraduate education. </think>
<search> [{"query": "Search for the undergraduate school of Ang Lee", "intent": 1}] </search>
**Rewritten Queries:**

---

*Continued on next page*

Table 8: Prompt for agent-based CoT trajectory construction.

{"queries": ["Ang Lee education background", "Ang Lee undergraduate school", "Ang Lee biography"]}

**Content:** {input}
**Rewritten Queries:**

---

**Observer Agent**

---

Given a multi-hop search problem, the steps already taken, and the search results obtained from the last executed step, your task is to:
1. Carefully analyze the search results to determine if they resolve the previous step, and provide a summary.
2. Update the execution steps and propose new steps based on the search results. You may encounter the following scenarios:
    a. If the current search results are sufficient to arrive at the Final Answer to the problem, provide the final answer directly. Use the format: "`<think>` {thought process} `</think>` `<answer>` {answer} `</answer>` ". Please strictly adhere to this format.
    b. If the current search results resolve the previous step, proceed according to the original plan and provide the next new step. Use the format: "`<think>` {thought process} `</think>` `<search>` [{"query": "query", "intent": 1}] `</search>` "; the query should be in sentence format.
    c. If the current search results do not resolve the previous step, reflect on and update the previous execution steps based on the search results. Use the format: "`<think>` {reflection} `</think>` `<search>` [{"query": "query", "intent": 1}] `</search>` "; the query should be in sentence format.
    d. If after multiple reflections (at least twice), the search still fails to resolve the issue, supplement based on your knowledge and provide the next new step. Use the format: "`<think>` {thought process} `</think>` `<search>` [{"query": "query", "intent": 1}] `</search>` "; the query should be in sentence format.

Below are some examples:

**Question:** What is the undergraduate school of the director of the movie "Sense and Sensibility"?
**Thought:** `<think>` To find out the undergraduate school of the director of the movie "Sense and Sensibility", I will take the following steps:
1. First, determine who directed the movie "Sense and Sensibility".
2. Then, search for this director's educational background, focusing on undergraduate education. I will now begin with step one by searching online to identify the director of the movie "Sense and Sensibility". `</think>`
`<search>` [{"query": "Find out who directed the movie 'Sense and Sensibility'", "intent": 1}] `</search>`
**Observation:** `<information>`
1. The movie "Sense and Sensibility" was released in 1995 and is directed by Ang Lee, a renowned director known for his work in both Eastern and Western cinema.
2. Ang Lee is recognized for directing major films such as "Crouching Tiger, Hidden Dragon" and "Life of Pi".
3. Ang Lee's background in filmmaking and storytelling has earned him multiple Academy Awards and international acclaim. `</information>`
**Thought:** `<think>` After a detailed analysis of the search results, I concluded that Ang Lee is the director of the movie "Sense and Sensibility". Thus, I will proceed to the next step and search for Ang Lee's educational background, focusing on his undergraduate education. `</think>`
`<search>` [{"query": "Search for the undergraduate school of Ang Lee", "intent": 1}] `</search>`

**Question:** What is the undergraduate school of the director of the movie "Sense and Sensibility"?

*Continued on next page*

Table 8: Prompt for agent-based CoT trajectory construction.

**Thought:** `<think>` To find out the undergraduate school of the director of the movie "Sense and Sensibility", I will take the following steps:
1. First, determine who directed the movie "Sense and Sensibility".
2. Then, search for this director's educational background, focusing on undergraduate education. I will now begin with step one by searching online to identify the director of the movie "Sense and Sensibility". `</think>`
`<search>` [{"query": "Find out who directed the movie "Sense and Sensibility", "intent": 1}] `</search>`
**Thought**: `<think>` After a detailed analysis of the search results, I concluded that Ang Lee is the director of the movie "Sense and Sensibility". Thus, I will proceed to the next step and search for Ang Lee's educational background, focusing on his undergraduate education. `</think>`
`<search>` [{"query": "Search for the undergraduate school of Ang Lee", "intent": 1}] `</search>` **Observation**: `<information>`
1. Ang Lee graduated from the National Taiwan College of Arts in 1975 and then came to the U.S. to receive a B.F.A. Degree in Theatre/Theater Direction at the University of Illinois at Urbana-Champaign, and a Masters Degree in Film Production at New York University.
2. After high school Lee attended the Taiwan Academy of Art, where he became interested in acting. He moved to the United States in 1978 to study theater at the University of Illinois at Urbana-Champaign. `</information>`
**Thought**: `<think>` After a thorough analysis of the search results, I found out that Ang Lee graduated from the National Taiwan University with a degree in French literature, as well as a B.F.A. Degree from the University of Illinois at Urbana-Champaign, therefore, the answer is `</think>` `<answer>` National Taiwan College of Arts and University of Illinois at Urbana-Champaign. `</answer>`

**Question**: Who is the director of the suspense drama that premiered on MBC?
**Thought**: `<think>` To solve this problem, I need to follow a few steps in my search:
1. First, I need to identify which suspense drama premiered on MBC.
2. Then, I will look up who directed that suspense drama.
First, I will search for recent suspense dramas that premiered on MBC. `</think>`
`<search>` [{ "query": "What recent suspense dramas premiered on MBC?", "intent": 1}] `</search>`
**Observation**: `<information>` 1. Premiering soon! A 12-episode suspense drama is coming, starring Wang Zhiwen and Yu Ai... From the trailer, 'The Vanishing Elephant' combines suspense and comedy, bringing an amusing story with humor and absurdity.
2. Chen Weiting's suspense drama is airing, featuring time travel and criminal investigation, with top actors like award-winning actors joining."
3. The best 10 suspense dramas of the past 4 years, with 'The Big Squad' at the bottom. Though the series is framed as a historical drama, it unfolds as a large-scale suspense espionage drama.
4. Premiering soon! An 18-episode suspense drama is coming, starring Sun Li, with a strong cast. However, recent suspense dramas such as 'Dim Flames' and 'Rebirth' have not been well-received.
5. Lee Jong-suk's return! New drama 'The Black Lawyer' premiered on MBC, achieving high viewership. The story is a suspenseful crime drama about a vulnerable lawyer who is framed and battles powerful forces.
6. The top 10 suspense dramas to watch in 2024, featuring leading actors like Liao Fan, Duan Yihong, and Zhang Songwen...
7. Premiering tonight! Another suspenseful drama is coming! 'Thirteen Years in Dust' will air on iQIYI's Mist Theater, starring Chen Jianbin and Chen Xiao.
8. Chen Weiting's new drama 'The Sun and Stars' debuted with impressive suspenseful plotlines. `</information>`
**Thought**: `<think>` From the search results, I found several relevant pieces of information:
1. The suspense drama 'The Black Lawyer' premiered on MBC, starring Lee Jong-suk and Im Yoon-ah, and it is a crime suspense drama.
2. The premiere of 'The Black Lawyer' achieved high ratings, telling a story about a vulnerable lawyer who is framed and fights back against powerful figures.

*Continued on next page*

Table 8: Prompt for agent-based CoT trajectory construction.

3. However, the search results did not provide the director's information for 'The Black Lawyer.' Filter out unrelated content: The search results also mention suspense dramas on other platforms like iQIYI and Mist Theater, and other non-MBC dramas, such as 'The Sun and Stars.' These are not directly related to the question and can be excluded.
Current progress summary: Based on the retrieved information, 'The Black Lawyer' is a recent suspense drama premiered on MBC, but the director's information is still missing.
Next plan: I will continue with the second step of my plan, searching for the director of 'The Black Lawyer.' `</think>`
`<search>` [{"query": "Who is the director of 'The Black Lawyer'?", "intent": 1}] `</search>`
**Question:** {input}
**Thought:** {thought}
**Observation:** {observation}
**Thought:**

---

**LLM-Judge**

Given the correct answer to a question, determine if the modelś response is correct. If correct, output "correct"; if incorrect, output "incorrect". Do not include unrelated content.

**Question:** {question}
**Correct Answer:** {answer}
**Model Response:** {model_response}

---

# G    REINFORCEMENT LEARNING

## G.1    TRAINING TEMPLATE

Table 9: Training template. The question is appended at the end during RL training and inference.

Answer the given question. You must conduct reasoning inside `<think>` and `</think>` first every time you get new information. After reasoning, if you find you lack some knowledge, you can call a search engine by `<search>` query `</search>`, and it will return the top searched results between `<information>` and `</information>`. You can search as many times as you want. If you find no further external knowledge needed, you can directly provide the answer inside `<answer>` and `</answer>` without detailed illustrations. For example, `<answer>` xxx `</answer>`. Question:

As illustrated in Table 9, during the RL process, we follow Jin et al. (2025) by utilizing a multi-round interactive template to guide the reasoning of the policy model. Specifically, the model engages in internal reasoning within the ¡think¿ tag, where it analyzes the problem and assesses the information collected. If additional evidence is required, the search query is refined within the ¡search¿ tag. Once sufficient information is gathered, the answer is provided within the ¡answer¿ tag.

## G.2    RL ALGORITHMS DETAIL

**DAPO with Search Engine**    Decouple Clip and Dynamic sAmpling Policy Optimization (DAPO) Yu et al. (2025) is an advanced RL algorithm that enhances the Group Relative Policy Optimization (GRPO) Shao et al. (2024) by incorporating techniques such as dynamic sampling and token-level policy gradient loss. DAPO samples a group of outputs $\{o_i\}_{i=1}^{G}$ for each question $q$ paired with the answer $a$, and optimizes the policy model via the following objective function:

$$\mathcal{J}_{\text{DAPO}}(\theta) = \mathbb{E}_{(q,a)\sim\mathcal{D},\{o_i\}_{i=1}^{G}\sim\pi_{\theta_{\text{old}}}(\cdot|q)}$$

$$\left[ \frac{1}{\sum_{i=1}^{G}|o_i|} \sum_{i=1}^{G}\sum_{t=1}^{|o_i|} \min\left( r_{i,t}(\theta)\hat{A}_{i,t}, \text{clip}\left(r_{i,t}(\theta), 1-\varepsilon_{\text{low}}, 1+\varepsilon_{\text{high}}\right)\hat{A}_{i,t} \right) \right] \quad (8)$$

where

$$r_{i,t}(\theta) = \frac{\pi_\theta(o_{i,t} \mid q, o_{i,<t}; \mathcal{R})}{\pi_{\theta_{\text{old}}}(o_{i,t} \mid q, o_{i,<t}); \mathcal{R})}, \quad \hat{A}_{i,t} = \frac{R_i - \text{mean}(\{R_i\}_{i=1}^G)}{\text{std}(\{R_i\}_{i=1}^G)}. \tag{9}$$

Here, $\mathcal{R}$ is the search engine, $\hat{A}_{i,t}$ represent the advantage, $\varepsilon_{\text{low}}$ and $\varepsilon_{\text{high}}$ are hyperparameters where we set 0.2 and 0.28 respectively. Additionally, DAPO removes KL Divergence to stabilize the generation of long-cot. We mask the retrieved tokens from the search engine, ensuring the policy gradient objective is computed only over LLM-generated tokens.

## G.3 Model-based Reward Prompt

Table 10: Prompt for model-based reward design.

**Reward Model**

Please evaluate whether the model's response is correct based on the given question, standard answer, and the model's predicted answer. Your task is to rate the result as: **Correct** or **Incorrect**.

Correct Responses

Here are examples of **Correct** responses:
Question: What are Barack Obama's children's names?
Standard Answer: Malia Obama and Sasha Obama
Model Prediction 1: Malia Obama and Sasha Obama
Model Prediction 2: Malia and Sasha
Model Prediction 3: Most people would say Malia and Sasha, but I'm not sure and need to confirm.
Model Prediction 4: Barack Obama has two daughters, Malia Ann and Natasha Marian, but they are commonly known as Malia Obama and Sasha Obama. These responses are **Correct** because:
They fully include the important information from the standard answer.
They do not contain any information that contradicts the standard answer.
Only the semantic content is considered; language (English or Chinese), case, punctuation, grammar, and order are not important.
The presence of vague statements or guesses is acceptable, as long as the standard answer is included and there is no incorrect or contradictory information.

Incorrect Responses

Here are examples of **Incorrect** responses: Question: What are Barack Obama's children's names?
Standard Answer: Malia Obama and Sasha Obama
Model Prediction 1: Malia
Model Prediction 2: Malia, Sasha, Susan, and Sasha Obama or Malia Obama, or Natasha Marian, or Einstein
Model Prediction 3: Although I don't know their exact names, I can say that Barack Obama has two children.
Model Prediction 4: You might be thinking of Bessie and Olivia. But you should check the latest references for detailed information. Is that the correct answer?
Model Prediction 5: Barack Obama's children These responses are **Incorrect** because: They contain factual statements that contradict the standard answer.
The answer is empty, restates the question.
The answer lists multiple answers, restates the answer.

Special Notes

Please note the following:
The standard answer may contain multiple aspects of the question's response, and within the same aspect, there may be multiple different descriptions, all of which are correct and are given within the same parentheses, connected by commas. For example, consider the question "What is the name of the social media platforms purchased by Elon Musk?":
Predicted answers "Twitter," "Twitter, X," and "X" are all **Correct**.

*Continued on next page*

Table 10: Prompt for model-based reward design.

For standard answers that contain responses to multiple aspects of the question, the model must provide answers to all aspects to be considered correct; otherwise, it is directly judged as **Incorrect**. There is no such output as **Partially Correct**. These answers will be given in different parentheses. For example, consider the question "Who are the original members of the band The Beatles?":

Predicted answers "John Lennon, Paul McCartney, George Harrison, Ringo Starr" that include all answers are considered **Correct**.

Predicted answers like "John Lennon, Paul McCartney" that do not include all answers are considered **Incorrect**.

ADDITIONAL GUIDELINES

Also, pay special attention to the following:

For questions with numerical standard answers, the predicted answer should match the standard answer. For example, consider the question "What is the total length of the Jinshan Railway Huangpujiang Special Bridge in meters?":

Predicted answers "3518," "3518.1," and "3518.17" are all **Correct**.

Predicted answers "3520" and "3600" are all **Incorrect**.

If the model's prediction does not directly answer the question and attempts to bypass or fails to directly provide the standard answer, it is considered an **Incorrect** answer.

If the standard answer contains more information than the question, the predicted answer only needs to include the information mentioned in the question.

If it is obvious from the question that the predicted answer has omitted information, it is considered **Correct**.

If it is clear that different translation versions of a name refer to the same person, they are also considered **Correct**.

You should focus more on the match between the standard answer and the model's prediction, rather than whether the standard answer is correct.

EXAMPLE QUESTION

Here is a new example question. Please rate the predicted answer as one of the following:
Question: {question}
Standard Answer: {target}
Predicted Answer: {predicted answer}
Only return the option represented by **Correct** or **Incorrect**, that is, only return A or B, without adding any other text.

### G.4 STABILITY ANALYSIS OF RL TRAINING STAGES AND REWARD MECHANISMS

We conducted an analysis of the stability across different stages and methods during Reinforcement Learning (RL) training.

#### G.4.1 COMPARATIVE ANALYSIS OF REWARD DESIGNS

We perform a comparative analysis of policy entropy during RL training under various reward mechanisms. As illustrated in the Figure 7, under the Model-based Reward mechanism, the entropy value initially increases and then stabilizes as training progresses. This trend is a strong indicator of a stable training process that successfully avoids policy collapse (both sudden collapse and explosive growth).In contrast, under the "Token-based Recall Reward" mechanism, we observed that the entropy value increases rapidly and exhibits abnormal behavior. We speculate that this is directly related to the reward hacking and the unexpected increase in response length that we analyzed previously. Furthermore, the entropy based on the Penalty Reward also demonstrates a similarly unstable state. These comparative trends collectively underscore the advantages and superior stability of the Model-based Reward mechanism.

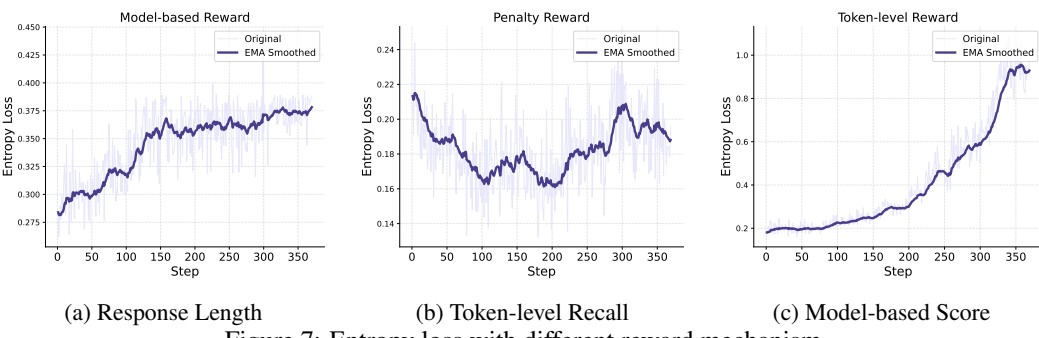

(a) Response Length        (b) Token-level Recall        (c) Model-based Score

Figure 7: Entropy loss with different reward mechanism

### G.4.2    STABILITY FOR PRE-TRAINING AND POST-TRAINING TASKS

We also provide detailed plots tracking Entropy, Response Length, and Reward Score for both the Pre-Training stage in Figure 8 and the Post-Training RL stage in Figure 9.

Pre-Training Stage: We observe a smooth and stable progression. The response length gradually increases and stabilizes around 2,500 tokens, while the reward score consistently improves. Crucially, the policy entropy shows a gradual increase followed by a period of sustained stability.

Post-Training Stage: A similarly healthy and steady trend is observed, with the policy entropy maintaining a constant, stable level. In both phases, the gradual rise and subsequent stability in entropy (rather than a sudden crash or sharp spike) confirms that our RL process is highly stable and effectively avoids both policy collapse and gradient explosion. This demonstrates the robust convergence and stability of our proposed method.

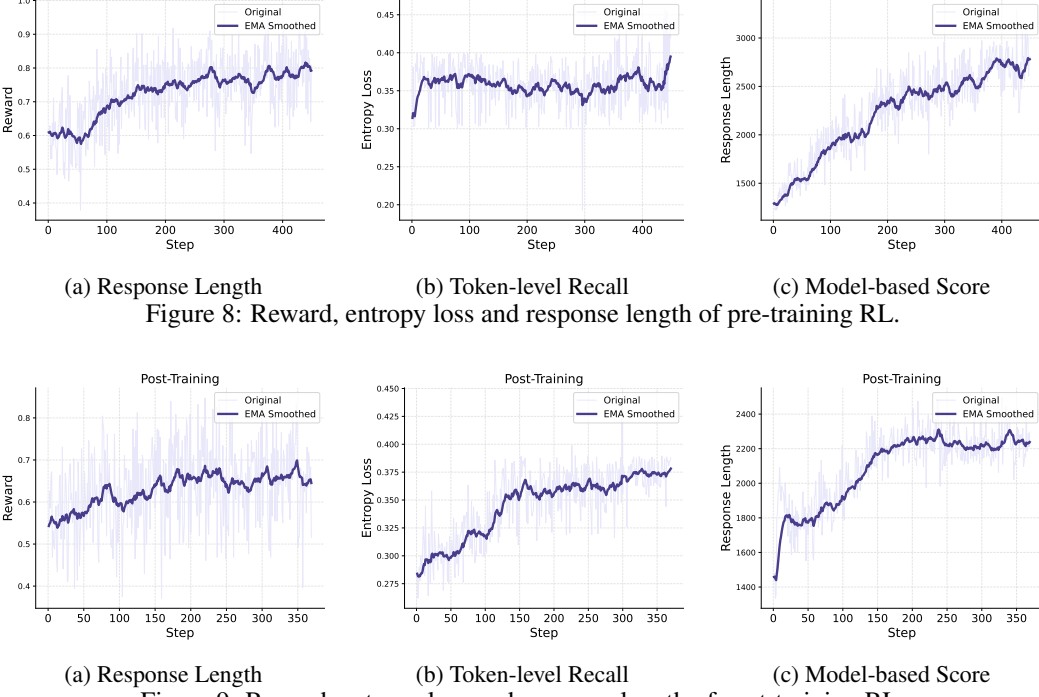

(a) Response Length        (b) Token-level Recall        (c) Model-based Score

Figure 8: Reward, entropy loss and response length of pre-training RL.

(a) Response Length        (b) Token-level Recall        (c) Model-based Score

Figure 9: Reward, entropy loss and response length of post-training RL.

## H    RAMP CASE STUDY

We illustrate how the model acquires the ability to retrieve and reason using the Retrieval-Augmented Mask Prediction Task (RAMP) through a case study presented in Table 11. In this example, information about the league, team, and university that David Henry Hoelscher is associated with is masked. The model autonomously utilizes the search engine to reason and fill in the missing information.

Initially, the model establishes a search plan, which includes details about the individual's career, team, and university. In the first step, it retrieves information related to his career and team. After acquiring the initial search results, the model identifies the league associated with the individual based on the retrieved data and initiates another search to confirm the team he played for. Following the second search, the model successfully infers his team affiliation and discovers that the previous search results include his college information. It then requests another search to verify this data. Upon completing the third search, the search agent effectively gathers all required information and fills in the masks.

Through the RAMP task, the model not only learns to perform retrievals but also masters the use of retrieved information for further reasoning and verification until the complete answer is obtained.

Table 11: RAMP Case.

| Question |
| --- |
| David Hoelscher David Henry Hoelscher (born November 27, 1975) is a former American football defensive tackle. He played one game in the [mask] for the [mask] in 1998. He played college football at [mask].
Fill in all the [mask] and output the whole paragraph without changing its format. |

| Response |
| --- |
| `<think>` To fill in the blanks in the provided sentence about David Hoelscher, I will perform the following searches:
1. First, search for David Hoelscher's brief NFL career to determine the specific league and team he played for in 1998.
2. Then confirm the name of the college where he played football before entering the NFL.
Now, I will start with the first step and search for David Hoelscher's NFL career details.
`</think>`
`<search>` [{"query": "David Hoelscher one game NFL" "David Henry Hoelscher NFL team 1998" "David Hoelscher defensive tackle one game in 1998", "intent": 1}]
`</search>`
`<information>`
1. David Hoelscher, , - Career Stats - NFL - CBSSports.com: Get the latest career stats for David Hoelscher of the on CBS Sports.
2. David Hoelscher, , - 2021-22 Game Log - NFL - CBSSports.com: Get up-to-date stats for every game played by David Hoelscher during the 2021-22 NFL season on CBS Sports.
3. David Hoelscher Stats, News and Video - DE — NFL.com: David Hoelscher DE Player Info Height 6-6 Weight 261 Arms Hands Experience 2 College Eastern Kentucky.
4. David Hoelscher - 2021-22 - Esports - Florida Southern College: David Hoelscher. Class Freshman. Hometown Orchard Park, N.Y.
5. David Hoelscher — American Football Database - Fandom: American football defensive tackle in the National Football League for the Washington Redskins. He played college football at Eastern Kentucky University.
`</information>`
`<think>` After finding out that David Hoelscher is indeed associated with the National Football League (NFL) and a specific team, I will now confirm the name of the team he played for in 1998 when he played one game.
`</think>`
`<search>` [{"query": "David Hoelscher one game in 1998 NFL team" "Which team did David Hoelscher play for in one game in 1998" "David Hoelscher NFL team when he played college football at Eastern Kentucky", "intent": 1}]
`</search>`
`<information>`
1. David Hoelscher 1998 Game Log — StatMuse: David Hoelscher's game log for the 1998 NFL regular season and playoffs. TEAM, W, L, T, PCT. Eagles NFL 2024 Betting. Best Record ATS · Chargers. 12-5-1.
2. David Hoelscher Stats, News and Video - DE — NFL.com: David Hoelscher DE Player Info Height 6-6 Weight 261 Arms Hands Experience 2 College Eastern Kentucky. |

Table 11: RAMP Case.

3. David Hoelscher - Washington Redskins Defensive Lineman: David Hoelscher played 1 season for the Redskins. He played but didn't record a tackle.
4. Colonels in the NFL - Eastern Kentucky University Athletics: Colonels in the NFL; David Hoelscher (1994-97), Washington Redskins (1998); Joe Hollingsworth (1946-47), Pittsburgh Steelers (1949-51); Tyrone
5. David Hoelscher Pro Football Stats, Position, College, Draft: DAVID HOELSCHER; 1998 Washington Redskins (NFL), 91, DT, 1, 0; 1999 Frankfurt Galaxy (NFLE), 91, RDT-RDE.
`</information>`
`<think>` After confirming that David Hoelscher played for the Washington Redskins in 1998, I will now verify if there is any additional confirmation needed about the college he attended before entering the NFL, which was mentioned as Eastern Kentucky in the previous searches.
`</think>`
`<search>` [{"query": "David Hoelscher college football" "Which college did David Hoelscher play football before entering the NFL?" "David Hoelscher defensive tackle college", "intent": 1}]
`</search>`
`<information>`
1. David Hoelscher Pro Football Stats, Position, College, Draft: DAVID HOELSCHER. Hoelscher, David Henry Height: 6-6 Weight: 256. Born: November 27, 1975 Coldwater, OH High School: Versailles (OH). Year, College, Status. 1994.
2. HOELSCHER; 1994, Eastern Kentucky, Roster; 1995, Eastern Kentucky, Roster; 1996, Eastern Kentucky, Roster; 1997, Eastern Kentucky, Played.
3. David Hoelscher - 2021-22 - Esports - Florida Southern College: David Hoelscher. Class Freshman. Hometown Orchard Park, N.Y.. Prev School Athletics. 111 Lake Hollingsworth Drive — Lakeland FL 33801 — 863-680-4245.
4.David Hoelscher Stats, Height, Weight, Position, Draft, College: Position: DE; Born: November 27, 1975 in Coldwater, OH; College: Eastern Kentucky; Weighted Career AV (100-95-): 0 (17977th overall since 1960); High School:.
5. David Hoelscher — American Football Database - Fandom: He played college football at Eastern Kentucky University. Stub icon, This biographical article relating to an American football defensive lineman born in the.
`</information>`
`<think>` After verifying the college where David Hoelscher played football, I can now complete the original statement.
`</think>`
`<answer>` David Hoelscher David Henry Hoelscher (born November 27, 1975) is a former American football defensive tackle. He played one game in the National Football League for the Washington Redskins in 1998. He played college football at Eastern Kentucky University.
`</answer>`

## I  FAILURE ANALYSIS AND CASE STUDIES

To provide a detailed understanding of the agent's limitations, we conducted both a quantitative and qualitative analysis of agent errors. We utilized a large language model (LLM) to automatically categorize failure instances across six distinct datasets into two primary error types:
1. F1 (Search Failure): The agent failed because it could not retrieve the required, relevant information from the search results, regardless of how many steps were executed.
2. F2 (Reasoning Failure): The agent failed despite the necessary, relevant information being present within the retrieved search results. The error was due to incorrect synthesis, selection, reasoning, or hallucination based on the available data.

| Failure Type | HotpotQA | FanoutQA | Musique | 2Wiki | Bamboogle | FreshQA |
|---|---|---|---|---|---|---|
| F1 | 55.20% | 32.44% | 61.50% | 30.25% | 51.85% | 33.01% |
| F2 | 44.80% | 67.56% | 38.50% | 69.75% | 48.14% | 66.98% |

Table 12: Quantitative Analysis of Agent Failure Modes

The breakdown of failure types across the evaluated datasets is presented in the Table **??**. This analysis reveals a significant task-dependence in the cause of failure. For datasets like Musique and HotpotQA, the primary bottleneck is Search Failure (F1), with these tasks requiring more complex, multi-step information retrieval that the search module struggles to satisfy. Conversely, for datasets such as FanoutQA, 2Wiki, and FreshQA, Reasoning Failure (F2) is the dominant error mode, accounting for over 66% of total failures. This strongly indicates that simply finding the correct documents is insufficient; the critical challenge lies in the agent's ability to correctly synthesize, select, and reason over potentially conflicting or complex retrieved information.

We provide two representative case studies that illustrate these distinct failure modes:

Case Study 1: F1 - Search Failure

Table 13 demonstrates an F1 (Search Failure) case, where the agent fails to find the key information and fabricates an answer based on irrelevant data. In this Trajectory: 1. Success: The agent correctly identified the author as Carlo Rovelli. 2. Failure (Search): The search failed to find the specific year he started working in France. The correct answer ("2000") was absent from all retrieved search snippets. 3. Failure (Reasoning): Instead of re-searching, the agent hallucinated a timeline. It incorrectly used Rovelli's 1986 PhD graduation date (which occurred in Italy) to falsely conclude he began working in France "around 1986," fabricating an answer that was not supported by the search results.

Table 13: F1 Case.

| **Question** |
| --- |
| Seven Brief Lessons on Physics was written by an Italian physicist that has worked in France since what year? |

| **Ground Truth Answer** |
| --- |
| 2000 |

| **Response** |
| --- |
| `<think>` To answer this question, I will perform the following searches: 1. First, identify the Italian physicist who authored "Seven Brief Lessons on Physics." 2. Next, determine the specific year when this physicist began working in France after moving there from Italy. Starting with the first step, I'll search for the Italian physicist associated with the book "Seven Brief Lessons on Physics. `</think>` `<search>` {"queries": ["Seven Brief Lessons on Physics author Italian physicist", "Italian physicist famous for writing Seven Brief Lessons on Physics about physics"]}] `</search>` `<information>` Seven Brief Lessons on Physics by Carlo Rovelli - Goodreads – Carlo Rovelli is an Italian theoretical physicist and writer who has worked in Italy and the USA, and currently works in France. His work is mainly in the Seven Brief Lessons on Physics by Carlo Rovelli - Goodreads – Carlo Rovelli is an Italian theoretical physicist and writer who has worked in Italy and the USA, and currently works in France. His work is mainly in the field Seven Brief Lessons on Physics - vialogue – Carlo Rovelli is a theoretical physicist who has made significant contributions to the physics of space and time. He has worked in Italy and the US, Seven Brief Lessons on Physics - vialogue – An intriguing one, for instance, is the mathematical theory being developed by Giulio Tononi – an. Italian scientist working in the United States. It's Seven brief lessons on physics - Carlo Rovelli @ 5x15 - YouTube – Carlo Rovelli is an Italian theoretical physicist and writer who has worked worldwide in the US, UK and Italy, and is currently the head of |

Table 13: F1 Case.

Seven Brief Lessons on Physics by Carlo Rovelli review – Born of a series of articles in an Italian newspaper and covering just seven topics, Rovelli's book conveys a simple truth: physics is Seven Brief Lessons on Physics: Rovelli, Carlo - Amazon.com – Italian theoretical physicist and writer Carlo Rovelli uses a conversational tone to untangle the most complicated yet most beautiful advances in science in Book Review — Seven Brief Lessons on Physics by Carlo Rovelli – Carlo, who published his book in 2014, is an Italian theoretical physicist. The author, an accomplished scientist himself, remarked Book Review — Seven Brief Lessons on Physics by Carlo Rovelli - This afternoon, I finished Seven Brief Lessons on Physics by Carlo Rovelli. Carlo, who published his book in 2014, is an Italian theoretical </information>
<think> After confirming that the author is indeed Carlo Rovelli, the next step is to confirm the specific year when he moved to France permanently or started his notable career there after leaving Italy. Therefore, the second part of the query needs to be addressed.
</think>
<search> {"queries": ["Carlo Rovelli first worked in France after leaving Italy", "When did Carlo Rovelli become a recognized physicist in France following his career in Italy?"]}
</search>
<information> Carlo Rovelli in conversation - Idler – snippet:Quantum physicist Carlo Rovelli's best-known work is Seven Brief Lessons On Physics, which has sold a million copies and been translated into Carlo Rovelli - High Profiles – Carlo Rovelli is a theoretical physicist and best-selling author who has won a place 'among the pantheon of great scientist-communicators' (New Scientist) Carlo Rovelli – In 1981, Rovelli graduated with a BS/MS in physics from the University of Bologna, and in 1986 he obtained his PhD at the University of Padova, Italy. Rovelli Carlo Rovelli – Carlo Rovelli (born 3 May 1956) is an Italian theoretical physicist and writer who has worked in Italy, the United States, France, and Canada. The Physics and Philosophy of Time - with Carlo Rovelli - YouTube – From Boltzmann to quantum theory, from Einstein to loop quantum gravity, our understanding of time has been undergoing radical The Order of Time: Amazon.co.uk: Rovelli, Carlo – He is a theoretical physicist, born in Italy but now working in France in the field of quantum-loop gravity - one of the contenders for the 'Theory of Carlo Rovelli - Berggruen Institute – In 1981, Rovelli graduated with a BS/MS in physics from the University of Bologna, and in 1986 he obtained his PhD at the University of Padova, Italy. He held Carlo Rovelli: 'Time travel is just what we do every day...'– The theoretical physicist and bestselling author answers questions from famous fans and Observer readers University welcomes Italian physicist Carlo Rovelli - Italian Carlo Rovelli is one of the pioneers in research on quantum gravity. He graduated in Physics from University of Bologna, with a doctorate and post- Carlo Rovelli on his search for the theory of everythin– Rovelli's passion for physics came by an unusual route: political activism. Born in 1956, he grew up in Verona, a provincial Italian town where" </information>
<think> After confirming both pieces of information, we confirm the complete answer. Carlo Rovelli is confirmed as the Italian physicist referenced. He completed his studies in Italy but has had a distinguished career internationally. Therefore, after verifying his professional trajectory, it is clear that shortly after gaining recognition beyond academia, he established himself in France, where he is noted for his contributions to theoretical physics, particularly quantum-loop gravity, which aligns him with academic institutions outside Italy. Hence, Carlo Rovelli began his recognized career as a theoretical physicist in France shortly after earning his PhD in 1986. Thus, he has been associated with significant scientific roles in France since shortly after completing his formal education, placing the earliest notable mention of his career path towards France around the late 1980s onwards. Therefore, Carlo Rovelli started working prominently in France after obtaining his PhD in 1986, marking the beginning of his recognized tenure there following his Italian origins.
</think>

*Continued on next page*

Table 13: F1 Case.

<answer> The Italian physicist who wrote "Seven Brief Lessons on Physics" is Carlo Rovelli, a theoretical physicist. He has been associated with notable scientific institutions in France after moving there from Italy. Carlo Rovelli began his recognized career path towards France shortly after his academic achievements; therefore, we confirm that he has worked in France following his career progression after graduating from Italy around 1986 when he earned his PhD. However, for the most precise timeline regarding his career prominence in France, we note he has been identified as a leading physicist there since the early 1990s onwards after holding positions in various countries including France post-doctorate periods abroad. Therefore, Carlo Rovelli is the correct identification, aligning his noted career movements to confirm he has been professionally significant in France since shortly after his academic milestones placed him on the international scientific stage following his graduation dates provided. </answer>

Case Study 2: F2 - Reasoning Failure
Table 14 demonstrates an F2 (Reasoning Failure) case, where the agent successfully finds the correct information but makes an incorrect selection when faced with conflicting data. In this Trajectory: 1. Success (Search): The agent correctly identified the arena as the "Androscoggin Bank Colisée" and searched for its seating capacity. 2. Success (Data Retrieval): The search results successfully retrieved the correct answer. Both Snippet 4 ("seats 3,677 for hockey") and Snippet 8 ("3,677") contained the exact number from the ground truth. 3. Failure (Reasoning): The agent's search also returned conflicting numbers (e.g., 3,675 and 3,682). When faced with this conflicting data, the agent's reasoning incorrectly selected the wrong number ("3,675") for its final answer, even though the correct answer was present in its retrieved information. This highlights the importance of improving the Search Agent's ability to resolve conflicting information when multiple sources are available.

Table 14: F2 Case.

| Question |
| --- |
| The arena where the Lewiston Maineiacs played their home games can seat how many people? |

| Ground Truth Answer |
| --- |
| 3,677 seated |

| Response |
| --- |

<think> To answer this question, I will perform the following searches:
1. First, search for information about the Lewiston Maineiacs to determine the name of the arena they played in during their existence as an ice hockey team.
2. Next, search for details about this specific arena to find out its seating capacity.
Starting with the first step, I'll search for the Lewiston Maineiacs to identify the arena associated with them. </think>
<search> {"queries": ["Lewiston Maineiacs home arena", "Ice hockey arena where Lewiston Maineiacs played after their last season"]}
</search>
<information> Androscoggin Bank Colisee, Lewiston Maineiacs – the arena in which the Maineiacs played is still standing and capable. After the Maineiacs folded, the arena mostly sat empty until 2015
Lewiston Maineiacs - Wikipedia –The team played its home games at the Androscoggin Bank Colisée. They were the second QMJHL team in the United States, and the only one to play a full season
MAINEiacs Nation — Facebook – In your QMJHL rookie season you played for the Lewiston Mainiacs what was your thoughts playing for an American team in the QMJHL? I've got nothing but
f only Lewiston could be a hockey town again . . . - Sun Journal – On The Colisée ice, where the Maine Nordiques perform their magic, an early game brawl was just concluding and the scrapper from the other team
Maine Perspective, v 4, i 16 – the Maine Center for the Arts at 8 p.m., Jan. 23. The gala will feature an evening highlighted with performances by University faculty, students and friends

*Continued on next page*

Table 14: F2 Case.

Maine Perspective, v 2, i 35 – more seats for fans, the University of Maine is starting construction this spring on the. 3.8 million expansion to Alfond Arena. of the IACS for a two-year

Civic responsibility — The city of Lewiston tries to . . . . . . - Mainebiz – As general manager of the Lewiston Colisee, the 45-year-old hockey arena formerly known as the Central Maine Civic Center, Wayne Thornton's goal is to generate

Maine Hockey History – Lewiston MAINEiacs (2003-2011) Although not a professional team, the MAINEiacs competed in the Quebec Major Junior Hockey League (QMJHL), considered one of Maine Campus October 30 1998 - CORE – Maine-iacs to stir up the sagging support for the team. The Maine-iacs did their job beautifully last weekend. But instead of receiving a well-deserved pat

Clarkson Names JF Houle as Next Men's Hockey Coach – Houle would leave Potsdam in 2009 to take over as Head Coach with the Lewiston Maineiacs of the QMJHL. After the team folded, he was immediately </information>

<think> After confirming that the Lewiston Maineiacs played their home games at the Androscoggin Bank Colisée, I will now search for the seating capacity of this arena to complete the original statement about the Lewiston Maineiacs.
</think>

<search> {"queries": ["Androscoggin Bank Colisée seating capacity", "How many people can the former Lewiston Maineiacs home arena hold?"]]}
</search>

<information> Darryl Antonacci sees potential in Androscoggin Bank Colisee – The Colisee seats 3,682 for hockey events, according to the Colisee website, while the Johnstown Tomahawks play in the 4,000-seat Cambria

Androscoggin Bank Colisee, Lewiston Maineiacs – Capacity: 3,675. Built: 1958. Address: 190 Birch St., Lewiston The seating area consists of ten rows of seating in the four-corner

Androscoggin Bank Colisee, – The arena was built for, and remains the home of, the Lewiston and St. Dom's high school teams, but later hosted concerts, exhibitions by the

Androscoggin Bank Colisee - The Maine Sports Commission – The facility seats 3,677 for hockey and 4500+ for concerts and provides more than 700 parking spaces on-site.

The Colisee - Maine Nordiques - Stadium Journey – The Colisee is a multi-purpose arena that opened in 1958. Built to serve the local Lewiston/Auburn community, the arena has been the site of countless events

The Colisee - Stadium and Arena Visits – The Colisée in multiple places has the capacity as 4,000 with a seating capacity of 3,675. It seemed questionable looking at the seating

The Colisee - Stadium and Arena Visits – The Colisée in multiple places has the capacity as 4,000 with a seating capacity of 3,675. It seemed questionable looking at the seating chart

The Colisée - Wikipedia – The Colisée, formerly Androscoggin Bank Colisée, Central Maine Youth Center, Central Maine Civic Center, and Lewiston Colisee, is a 4,000 capacity (3,677

The Colisee: The Colisée - Home of Maine Nordiques – Colisee: Info@thecolisee.com The Colisée is a 67000 square foot multipurpose facility, seating approximately 2634 for hockey and up to 4500+ for concerts

Venue Info - Lewiston - The Colisee –The Colisée is a 67,000 square foot multipurpose facility, seating approximately 2,634 for hockey and up to 4,500+ for concerts. The multi-purpose facility can
</information>

<think> After confirming both pieces of information, we confirm the exact details requested. Therefore, the arena where the Lewiston Maineiacs played their home games is identified, and its current recognized seating capacity is confirmed. The Androscoggin Bank Colisée, now known simply as The Colisee or The Colisée, is a multi-purpose arena that was the home of the Lewiston Maineiacs during their tenure as a professional ice hockey team. This arena, with its historical ties to Lewiston, Maine, has a recognized seating capacity of 3,675 for hockey events when it served as the home of the Lewiston Maineiacs. Hence, the Lewiston Maineiacs' home arena could seat up to 3,675 people during their time there from 2003 to 2011 before the team folded due to financial issues.
</think>

Table 14: F2 Case.

---

`<answer>` The arena where the Lewiston Maineiacs played their home games is The Colisee (formerly known as the Androscoggin Bank Colisée), which has a seating capacity of 3,675 for hockey events. This confirms that the Lewiston Maineiacs' home arena could accommodate 3,675 spectators when they were an active team in the Quebec Major Junior Hockey League (QMJHL). `</answer>`

---

## J   BROADER IMPACTS

As an exploration in the field of Search Agents, MASKSEARCH has the following impacts:

- Provides a general pre-training method for Search Agent: MASKSEARCH introduces the Retrieval Augmented Mask Prediction Pre-training Task, pioneering the enhancement of an agent's retrieval and reasoning capabilities during the pre-training stage. This offers new perspectives for future developments in agent technology.
- Applications in Various RAG Fields: As a general pre-training task, it can be broadly applied to RAG applications across different knowledge domains.
- Negative social impacts: There are no negative social impacts foreseen.

## K   LIMITATIONS

Unlike prior works that aim at multi-tool or general-purpose agents, we specifically focus on the search agent, which is a fundamental and widely applicable agent form. Despite the empirical success and intuitive motivation of our approach, there are several limitations that warrant further investigation. First, we employ only a search tool for knowledge retrieval to adhere to the concept of RALM. We believe that the RAMP task could be generalized to incorporate the use of multiple tools. Future work could explore the diversity of the toolset, potentially expanding the application scope beyond open-domain QA to other scenarios. Furthermore, while our method has demonstrated promising results, a more in-depth theoretical analysis is necessary to understand the factors contributing to its effectiveness.

