# OpenReview forum: "MaskSearch: Towards Scalable Agentic Pre-Training for Search-Enhanced Reasoning"
_ICLR.cc/2026/Conference — Submitted to ICLR 2026_

### Official Review · Reviewer_GTez · 2025-10-30

**Soundness:** 3
**Presentation:** 2
**Contribution:** 2
**Rating:** 4
**Confidence:** 4

**Summary:**

This paper proposes MaskSearch, a two-stage training framework designed to enhance the search and reasoning capabilities of Large Language Models (LLMs) through scalable agentic pre-training. Existing Retrieval-Augmented Language Models (RALMs) often use passive retrieval, while training-based search agents are constrained by task-specific data. To address this, the authors introduce a novel pre-training task called RAMP.

Experimental results demonstrate that models pre-trained with RAMP show significant performance improvements on several open-domain multi-hop QA benchmarks compared to baselines trained only on the downstream task.

**Strengths:**

- This work reframes the training of agentic search and reasoning abilities as a self-supervised-like mask prediction task. RAMP can leverage massive unlabeled text corpora (like Wikipedia) to generate nearly infinite training data. This scalability is key to training more robust and general search agents.

- The experimental results demonstrate that adding the RAMP pre-training stage consistently and significantly outperforms baselines trained only with downstream task fine-tuning, across various model sizes (Qwen and LLaMA).

- The framework is not limited to a single training methodology. It supports both SFT and RL (DAPO) and provides an experimental comparison between them.

**Weaknesses:**

1. The "alignment" of the pre-training task with downstream tasks and its potential harm to generalization. The proposed RAMP task (Sec 2.2) is fundamentally a *Convergent* task, i.e., locating and reasoning about specific, masked facts. However, many real-world search tasks (e.g., "write a research report on...") are *Divergent*, requiring exploration, synthesis, and generation of new knowledge. Over-optimizing the RAMP task on search agents may lead to a "narrow-sighted" agent, harming its ability to handle divergent tasks.

2. The paper observes in Sec 4.2 (Fig 3) that although RAMP pre-training is effective, the performance improvement for the 7B model flattens as the data scale increases. As noted in the draft, this is likely due to an upper bound on the "self-evolved" data. This suggests that a quality bottleneck in SFT data limits the ceiling of the pre-training method, especially for bigger models.

3. The ablation study on masking strategies in Sec 5.1 shows that the harder PPL strategy does not always yield better results. This indicates that the core question of "what kind of RAMP task best promotes agentic learning" remains unanswered, and the current design may not be optimal.

**Questions:**

1. Have the authors tested the RAMP pre-trained model on "Wide Search" or "deep research" type benchmarks (such as those in arxiv:2506.11763 or 2508.07999)? If RAMP indeed harms performance on such tasks, this would be a limitation that should be discussed.

2.The diversity bottleneck of SFT data appears to limit the 7B model's performance. Have the authors tried using a stronger external model (instead of "self-evolution") to generate SFT trajectories, to test if the performance ceiling of the 7B model can be further unlocked by higher-quality data?

---

> ### Author Response · Authors · 2025-11-21
> **Response to Reviewer GTez**
>
> We appreciate the reviewer's constructive comments and suggestions. We have provided detailed responses to the raised concerns and incorporated the corresponding changes in the manuscript.
>
> **W1 & Q1: Generalization on Divergent Task**
> We appreciate the reviewer's concern that the convergent nature of the RAMP task might lead to a "narrow-sighted" agent and hinder generalization to divergent tasks.
>
> The RAMP pre-training task primarily focuses on building **foundational agentic capabilities** such as planning, search (tool-use), and reasoning, which are atomic skills applicable to both convergent (fact-based) and divergent (creative or generative) tasks. The specialization into convergence vs. divergence occurs during downstream fine-tuning, not during pre-training. Our work is scoped around multi-hop question answering to demonstrate the effectiveness of RAMP.
>
> To further address this concern, we conducted additional evaluations on the **DeepResearch Bench** (arxiv:2506.11763), which is designed to assess exploration and synthesis capabilities.
>
> Given our main experiments focus on the 7B scale, we compared two settings:
>
> 1. **RAMP + HotpotQA**: A model pre-trained on RAMP and then post-trained on HotpotQA.
> 2. **HotpotQA Only**: A baseline model post-trained on HotpotQA without RAMP pre-training.
>
> As shown in the table, the model pre-trained with RAMP outperforms the baseline across all metrics on the DeepResearch Bench. Most notably, while the **Overall** score increased by **2.06%**, the **Comprehensive** score saw a substantial improvement of **3.37%**. These results empirically demonstrate that rather than harming generalization, RAMP pre-training equips the agent with stronger capabilities to handle divergent tasks requiring information synthesis.
>
> | DeepResearch Bench | overall | comprehensiveness | insight | instruction_following | readability |
> |--------------------|---------|-------------------|---------|------------------------|-------------|
> | HotpotQA           | 15.75   | 11.68             | 7.08    | 25.90                  | 23.10       |
> | **RAMP + HotpotQA** | **17.87 (+2.06)** | **15.05 (+3.37)** | **7.44 (+0.36)** | **29.88 (+3.98)** | **24.27 (+1.17)** |
>
> **W2 & Q2: Bottleneck of SFT and Self-Evolve Data**
> We agree that the quality of training data is a key factor in the performance of RAMP pre-training via SFT. However, our results indicate that improving SFT alone is not the most effective path toward scaling agentic capabilities for larger models.
>
> In our setup, the 7B model saturates under SFT because the quality and diversity ceiling of the trajectories generated by another self-evolving 7B model is insufficient. Our experiment shows that RL optimizes agentic behaviors beyond what can be captured by supervised trajectories, demonstrating that it provides a more scalable and model-efficient way to unlock the capacity of larger models. Therefore, our contribution lies in demonstrating **how different training paradigms (SFT vs. RL) should be applied for different model sizes to effectively learn the RAMP task**.
>
> **W3: Ablation Study on Masking Strategies**
> Our ablation study on masking strategies indicates that a harder perplexity-based strategy, which is a useful method for MLM pre-training, does not always lead to improved results in our agentic pre-training.
>
> We highlight that this aims to offer a solution to this problem by demonstrating that masking strategies need to balance task difficulty with the model's ability to learn progressively. This suggests that the choice of masking strategy is non-trivial and plays a crucial role in the agent's learning process, leaving space for future exploration.

---

> > ### Author Response · Authors · 2025-11-27
> > **Kindly Reminder of our Response to Reviewer GTez**
> >
> > Dear Reviewer,
> >
> > We hope this message finds you well. We would love to ensure that we have addressed all your concerns satisfactorily. If there are any additional points or feedback you'd like us to consider, please let us know soon. Your insights are invaluable to us, and we are eager to address any remaining concerns to improve our work.
> >
> > Thank you for your effort in reviewing our paper and look forward to having further discussion with you.

---

> > ### Comment · Reviewer_GTez · 2025-11-28
> >
> > Thank you for your additional evaluation on the DeepResearch Bench. While the new results partially address my concerns regarding generalization to divergent tasks, I remain unconvinced about the significance of the core contribution.
> >
> > I **concur with** the points raised by Reviewer `uJ6v`. The proposed framework appears to be an engineering-oriented optimization of the standard SFT+RL pipeline rather than a fundamental paradigm shift. The theoretical justification for the RAMP task remains thin.
> >
> > Given these factors, I do not find sufficient grounds to raise my rating and will maintain my original score.

---

### Official Review · Reviewer_Ba3W · 2025-10-31

**Soundness:** 4
**Presentation:** 4
**Contribution:** 3
**Rating:** 8
**Confidence:** 3

**Summary:**

The paper introduces a pre-training task for LLMs specifically designed for search-oriented agents. The core idea is to define a mask-based prediction task that incorporates information retrieved through a search operation. In practice, a sentence is sampled from a dataset, and search-relevant information is masked. A solution trajectory is then generated via interaction with a search agent, and this trajectory is used to train the model through either supervised fine-tuning or RL-based methods. Finally, a post-training stage using multi-hop QA data is applied to enable question answering. The effectiveness of the approach is evaluated on two model families (LLaMA 3 and Qwen2.5) of various sizes and across multiple datasets.

**Strengths:**

1. The approach is novel. The idea of introducing a masked prediction pre-training task before applying the Search-R1-style post-training is smart and empirically shown to be effective.
2. The evaluation demonstrates consistent improvements across different architectures and model scales. The performance gains are significant and clearly support the usefulness of the method.
3. The manuscript is well written, with a clear problem setup and a well-presented methodological contribution.

**Weaknesses:**

While the authors have tested their method on multiple model architectures, it is unclear if such a pre-training strategy would scale effectively to larger models. This concern is particularly relevant given the observation, as shown in Figure 3, that performance improvements tend to saturate as model size increases. Although this is not a major issue and does not undermine the validity of the contribution, it would be helpful if the authors could comment on scalability toward larger models.

As a minor point, the paper could benefit from an analysis of the computational overhead introduced by the additional pre-training task. It would be useful to explicitly report how the computational cost of the search-based pre-training compares to that of the original pre-training and the QA-oriented post-training stages.

**Questions:**

1. Is there any intuition of the behavior on larger models?
2. What is the computational overhead?

---

> ### Author Response · Authors · 2025-11-21
> **Response to Reviewer Ba3W**
>
> We sincerely thank the reviewer for the encouraging remarks and acknowledging the significance of our work! We have carefully considered the comments and made improvements to further strengthen our manuscript.
>
> **W1: Scalability Toward Larger Models.**
> Thank you for the question. Our current experiments across 1.5B/3B/7B already provide strong evidence that MaskSearch is effective and stable across architectures and model sizes, and are aligned with the scale used in prior SOTA work (Search-R1, which uses 3B/7B models). Nothing in our method fundamentally limits such scaling, although we have not extended evaluation to larger models due to the computational cost of full-parameter training.
> The saturation trend in Fig. 3 primarily reflects the ceiling of SFT given the quality of supervised trajectories, rather than a limitation of MaskSearch. As also discussed in the paper, RL training continues to yield additional gains for larger models, indicating that the underlying search capability learned through RAMP pre-training remains scalable.
>
> **W2: Computational Cost.**
> We have added a concise comparison of the computational overhead introduced by the agentic pre-training stage in the revised manuscript. The results are additionally presented in **Appendix C**.
>
> (a) **Agentic Pre-training vs. Agentic Post-Training**: We summarize the wall-clock time for SFT and RL across Qwen model sizes in the following table and have included this table in the revised manuscript. The detailed training settings were described in Appendix B.
>
> | Model Size | Pre-training (SFT) | Pre-training (RL) | Post-training (SFT) | Post-training (RL) |
> |------------|--------------------|-------------------|---------------------|---------------------|
> | 7B         | 4 days 14 hours    | 5 days            | 11 hours            | 2 days              |
> | 3B         | 2 days 7 hours     | 3 days            | 8 hours             | 1 day 4 hours       |
> | 1.5B       | 22 hours           | < 2 days          | 6 hours             | 20 hours            |
>
> (b) **Agentic Pre-training vs. Original Pre-training**: The computational cost of pre-training the base model is not reported by Qwen [1], whereas it is reported that the pre-training dataset contains **18T** tokens. Our RAMP uses up to **25B** tokens, which is several orders of magnitude smaller. Thus, the computational cost of RAMP pre-training is much smaller compared to full pre-training.
>
> [1] Qwen2.5 Technical Report. https://arxiv.org/pdf/2412.15115.

---

### Official Review · Reviewer_uJ6v · 2025-11-01

**Soundness:** 2
**Presentation:** 2
**Contribution:** 2
**Rating:** 4
**Confidence:** 4

**Summary:**

The paper MASKSEARCH: Towards Scalable Agentic Pre-training for Search-Enhanced Reasoning introduces a unified two-stage training framework for large language models (LLMs) that integrates retrieval-based reasoning and agentic search abilities. The key innovation, Retrieval-Augmented Mask Prediction (RAMP), serves as a scalable pre-training task where the model learns to fill masked spans through multi-step search and reasoning, thereby simulating agentic behavior. The framework employs both Supervised Fine-tuning (SFT) and Reinforcement Learning (RL) in pre- and post-training stages, with a hybrid reward system and curriculum learning strategy. Extensive experiments across multiple open-domain multi-hop QA benchmarks demonstrate that MASKSEARCH substantially improves both in-domain and out-of-domain reasoning performance, effectively bridging foundational pre-training and search-augmented agentic learning.

**Strengths:**

1. Proposes the first scalable Retrieval-Augmented Mask Prediction (RAMP) task, connecting masked language modeling with agentic search reasoning.
2.  Elegantly combines pre-training and post-training using both SFT and RL, enhancing flexibility and generalizability.
3. Demonstrates consistent and transferable improvements across diverse multi-hop QA datasets and model sizes.
4. Integrates multi-agent data synthesis, self-evolution distillation, and curriculum learning to ensure scalability and stability.

**Weaknesses:**

1. The paper lacks a theoretical analysis of the convergence and generalization properties of the RAMP task, remaining primarily at the empirical level.
2. The framework shows strong structural similarities to prior works such as Search-R1 and DeepSeek-R1, making the boundary of innovation insufficiently clear.
3. Although multiple datasets are evaluated, the paper lacks fine-grained discussions on failure cases, cross-task transferability, and robustness to noise.
4. While the DAPO algorithm and reward design are described, the paper does not provide comparative analyses of training stability or convergence behavior.

**Questions:**

See the Weaknesses.

---

> ### Author Response · Authors · 2025-11-21
> **Response to Reviewer uJ6v (Part 1/3)**
>
> We appreciate the reviewer’s thoughtful comments and suggestions. We have provided detailed responses below and incorporated the corresponding changes in the manuscript.
>
> **W1: Theoretical Analysis of Convergence and Generalization Properties of RAMP.**
>
> Our work primarily focuses on demonstrating the empirical effectiveness and scalability of the RAMP task within a two-stage agentic training pipeline, as stated in the Limitation section (Appendix J). Nevertheless, we agree that providing theoretical intuition is valuable, and below we include a concise analysis:
>
> (a) **Convergence**: Under SFT, we minimize the token-level loss over full Chain-of-Thought trajectories $y_t$ produced by $\pi_t$: $\min_{\theta}\mathbb{E}_{(x, y_t)\sim D_t}\mathcal{L}(\pi _{\theta}(x, D_R), y_t))$. This objective is a token-level likelihood minimization over discrete outputs, and standard gradient-based optimization is known to converge to a first-order stationary solution under such conditions. Under RL, the DAPO objective is KL-regularized, ensuring bounded policy updates and guaranteeing convergence to a fixed point of the proximal policy optimization dynamics. Importantly, search results are masked during gradient computation to prevent instability arising from tool responses.
>
> (b) **Generalization**: RAMP reduces the conditional uncertainty of the prediction target by conditioning on search results $D_R$, effectively transforming the prediction problem from $H(y\mid x)$ to the much lower-entropy $H(y\mid x, D_R)$. This verifiability property is motivated by the pre-training of Masked Language Models and enables the agent to acquire transferable subskills—planning, tool calling, and multi-hop reasoning—that directly support downstream tasks involving search-enhanced reasoning, e.g. multi-hop QA.
>
> **W2: Difference with Prior Works**
>
> While previous works focus on RL algorithmic improvements for LLMs or search agents, MaskSearch introduces a **new training paradigm with agentic pre-training stage**:
>
> ● Search-R1/DeepSeek-R1: These methods apply RL directly on downstream tasks like (non-agentic) math or (agentic) multi-hop QA.
>
> ● MaskSearch: Our core contribution is the Retrieval-Augmented Mask Prediction (RAMP) task, which instills fundamental search and reasoning capabilities before the downstream task. We train with RL and SFT to validate the effectiveness of RAMP for search agents.

---

> ### Author Response · Authors · 2025-11-21
> **Response to Reviewer uJ6v (Part 2/3)**
>
> **W3: Failure Cases, Transferability, Robustness**
>
> ---
>
> (1) **Failure Cases**
>
> To address this, we have added a dedicated **Appendix H: Failure Analysis and Case Studies** in the revised manuscript, which details our quantitative and qualitative investigation of agent errors.
> Our analysis categorized failures into two primary types: **F1 (Search Failure)** and **F2 (Reasoning Failure)**.
>
> - **Quantitative Findings:**
>   The root cause of failure is task-dependent. For complex retrieval tasks like **Musique** and **HotpotQA**, the primary bottleneck is **F1 (Search Failure)**, where the necessary information is simply not retrieved. Conversely, for tasks like **2Wiki** and **FanoutQA**, **F2 (Reasoning Failure)** dominates, accounting for up to 70% of errors.
>
> - **Case Studies:**
>   This demonstrates that merely finding relevant documents is often insufficient. The critical challenge lies in the agent's ability to **synthesize, select, and resolve conflicting information** from the search results, as highlighted by a representative F2 case study where the agent incorrectly selected a conflicting capacity value despite the correct answer being available. An F1 case study further illustrates failure due to hallucination when search results are insufficient.
>
> These findings clarify future directions for improvement, emphasizing the need to enhance both the search module's precision and the agent's downstream reasoning capabilities.
> **We encourage the reviewer to examine the detailed data and case studies in Appendix H.**
>
> ---
>
>  (2) **Cross-Task Transferability**
>
> In our experiments, we specifically trained our model using HotpotQA as the in-domain dataset and evaluated its generalization capabilities on distinct out-of-domain datasets. To verify robustness, we tested on **FreshQA**, which introduces time-sensitive constraints (e.g., "What is the best-selling jazz album of all time?"), and **Bamboogle**, a handcrafted collection designed for information-seeking challenges (e.g., "What rocket was the first spacecraft that ever approached Uranus launched on?"). These results demonstrate that our method effectively transfers from standard multi-hop reasoning to both temporally dynamic and open-domain retrieval tasks.
>
> Furthermore, we extend our evaluation to the **DeepResearch Bench**, a benchmark dedicated to open-ended research report generation. We compared our RAMP-pretrained 7B model (post-trained on HotpotQA) against a baseline model without RAMP pre-training.  As shown in the Table, the RAMP-pretrained model consistently outperformed the baseline. Most notably, **it achieved a 2.06% increase in the Overall score and a substantial 3.37% improvement in the Comprehensive metric**. These results provide strong empirical evidence that the reasoning capabilities acquired during RAMP pre-training are not limited to convergent fact retrieval; instead, they significantly enhance the agent’s ability to explore, synthesize, and generate comprehensive content for complex, divergent tasks.
>
> | DeepResearch Bench | overall | comprehensiveness | insight | instruction_following | readability |
> |--------------------|---------|-------------------|---------|------------------------|-------------|
> | HotpotQA           | 15.75   | 11.68             | 7.08    | 25.90                  | 23.10       |
> | **RAMP + HotpotQA** | **17.87 (+2.06)** | **15.05 (+3.37)** | **7.44 (+0.36)** | **29.88 (+3.98)** | **24.27 (+1.17)** |
>
> ---
>
> (3) **Noise Robustness**
>
> All results in Tables 2–4 and Figure 3 are averaged over 3–5 independent runs with varying random seeds and real-life search engine responses, which accounts for noise that mainly comes from the volatility of LLMs and Search Engine Results Pages (SERPs). This robustness stems directly from RAMP pre-training, which trains the agent to recover from retrieval failures.
> We explicitly state this in the revised manuscript.

---

> ### Author Response · Authors · 2025-11-21
> **Response to Reviewer uJ6v (Part 3/3)**
>
> **W4: Comparative Analyses of Training Stability and Convergence**
>
> To comprehensively address this concern, we have added a new section, **Appendix F.4** in the revised manuscript. This section provides detailed empirical evidence demonstrating the robustness of our approach across different reward mechanisms and training stages.
>
> - **Stability of Reward Designs:** We compare the policy entropy trajectory under three reward mechanisms. Our analysis shows that the Model-based Reward mechanism results in a highly stable training process, where entropy initially rises and then stabilizes, successfully avoiding policy collapse or explosive behavior. In contrast, the "Token-based Recall Reward" and "Penalty Reward" exhibit unstable or abnormal entropy increases, which we attribute to issues like reward hacking. This strongly validates the superior stability of our chosen Model-based Reward.
>
> - **Stability Across Training Stages:** We further present detailed plots for the Pre-Training and Post-Training RL stages, tracking Entropy, Response Length, and Reward Score. In both stages, the policy entropy exhibits a gradual, controlled rise followed by sustained stability, confirming that our RL process is robust, avoids sudden crashes or spikes (policy collapse/gradient explosion), and demonstrates strong convergence.
>
> The data presented in Appendix F.4 confirms the high stability and robust convergence of our method. We encourage the reviewer to examine these detailed plots and analyses in the **revised manuscript's Appendix**.

---

> > ### Comment · Reviewer_uJ6v · 2025-11-22
> >
> > Thank you for the rebuttal, which partially addresses my concerns. However, I believe that there are already many DeepResearch-style approaches, and this method still relies on a slightly modified SFT+RL pipeline. It does not introduce a fundamental innovation or a paradigm-level advancement, but rather resembles an engineering-oriented optimization. In addition, the formal theoretical justification remains weak.

---

> > > ### Author Response · Authors · 2025-11-25
> > > **Response to Reviewer uJ6v on Innovation and Theoretical Analysis**
> > >
> > > Thank you for your timely and thoughtful feedback. We would like to further clarify our innovation and contribution, as well as the theoretical analysis.
> > >
> > > 1. **Approach and Paradigm Innovation**
> > >
> > >    We agree that many recent works on reasoning and agentic model share the same underlying SFT → RL training paradigm, including the baseline *Search-R1* we compare against, and many concurrent *DeepResearch-style* methods. These approaches **directly optimize the agent on the final downstream task**, e.g., DeepResearchBench, QA, and math. Their training objective is tied to the specific task distribution.
> > >
> > >    In contrast, **RAMP introduces a dedicated agentic pre-training stage that is absent in prior works**. This stage is task-agnostic and focuses on acquiring transferable foundational agentic abilities such as planning, retrieval, and search-enhanced reasoning before any task-specific supervision. To the best of our knowledge, our work is the **first** to formalize and operationalize such an *agentic pre-training* objective. This differs not in minor engineering details but in changing the training paradigm itself, i.e., learning agentic primitives prior to downstream optimization.
> > >
> > > 2. **More on Theoretical Justification**
> > >
> > >    It is acknowledged that concurrent DeepResearch-style works provide formal definition with primarily heuristic or empirical justification. Their contribution often arise from uncovering generalizable behaviors and principles, which cannot be derived from theory alone. Our focus is also on empirically grounded, paradigm-level insights.
> > >
> > >    **Moreover, in our previous response, we have presented both theoretical analysis and detailed empirical evidence for training stability, convergence and task generalization.** Hence, we would be grateful if the reviewer could indicate which specific component of our theoretical justification is viewed as insufficient, so that we can further strengthen the discussion in a focused manner.

---

### Author Response · Authors · 2025-12-01
**Rebuttal Summary**

For clarity, we refer to Reviewers uJ6v, Ba3W, and GTez as R1, R2, and R3, respectively, in the following response.

We sincerely thank all reviewers and ACs for their thoughtful consideration. We are encouraged by the recognition of our contributions and strengths, including the **novelty** of our approach (R2), the **scalability and consistent gains** (R1, R2, R3), and the **effective combination** of different training stages and methods (R1, R3).

We have also carefully addressed each individual comment and believe that we have responded to the majority of concerns with concrete analyses, empirical evidence, and manuscript revisions. Below, we summarize the core points raised and how they were addressed during the discussion phase, and the updates to our manuscript.

---

**1. Core Contribution and Difference with Prior Works (R1, R3)**: Our contribution lies in introducing **agentic pre-training**, a new training stage that learns transferable search and reasoning capabilities before any task-specific optimization, which materially changes the exsiting paradigm that directly performs training on downstream tasks, rather than making engineering refinements.

**2. Theoretical Justification (R1, R3)**: We provided **theoretical analysis for convergence and generalization** of RAMP, while emphasizing that our work is more **empirically grounded**, which is supported by additional experiments on stability, convergence, and generalization.

**3. Transferability and Generalization (R1, R3)**: We presented further evaluation on **DeepResearch Bench**, a benchmark for wide-search and synthesis abilities. RAMP-pretrained models consistently **outperform** the baseline across all metrics, including a **+3.37 improvement in Comprehensiveness**.

**4. Training Stability and Convergence (R1)**: We added detailed convergence curves (entropy, reward score, response length) across both pre-training and post-training RL stages.

**5. Failure Cases and Robustness (R1)**: We added a quantitative and qualitative analysis of **failure types** and detailed case analysis for each type. The robustness of our reported results is also clarified.

**6. Computational Overhead (R2)**: We presented a wall-clock cost analysis and clarified that RAMP uses orders of magnitude fewer tokens than base-model pre-training.

---

**Updates in Manuscript**:

**1. Section 6**: Further highlighting the difference of our work from prior agentic works.

**2. Appendix C**: Analysis of the computational cost introduced by the proposed agentic pre-training stage.

**3. Appendix G.4**: Analysis of the stability across different stages and methods during RL training.

**4. Appendix I**: Failure analysis and case studies.

---

We believe these additions and clarifications comprehensively address the concerns raised during the review process and substantially enhance the clarity and completeness of our manuscript. All revisions are clearly highlighted for ease of reference.

---

### Meta-Review · Area_Chair_x3BC · 2025-12-09

**Summary:**

The submission collects three review reports. The reviewers identify critical weaknesses that preclude acceptance.
1. The paper lacks theoretical analysis of RAMP’s convergence and generalization properties.
2. Its innovation boundary is unclear, with strong structural similarities to Search-R1 and DeepSeek-R1, appearing more as engineering optimization than fundamental advancement.
3. Scalability concerns are evident: performance improvements saturate with model size and encounter data quality bottlenecks, particularly for 7B models. Most fundamentally, R3 raises an architectural concern: RAMP’s convergent nature (locating masked facts) misaligns with divergent real-world search tasks requiring exploration and synthesis, risking “narrow-sighted” agents that harm generalization.

In addition, I also have concerns that the authors failed to show why using mask filiing task can enhance the reasoning ability. The study also lacks of theoretical temporal analysis.

Compounding this, the paper lacks comprehensive failure analysis, cross-task transferability studies, and robustness to noise The ablation revealing suboptimal masking strategies further suggests the RAMP design is not well understood.

**Reviewer Concerns:**

Addressed: Computational overhead analysis, additional scalability experiments, failure case studies, and clearer differentiation from prior work.

Outstanding: Fundamental theoretical analysis is a major research endeavor beyond rebuttal scope. The core innovation gap and the convergent-divergent task mismatch represent architectural flaws. The data quality bottleneck and suboptimal masking design indicate deeper methodological issues requiring substantial new experiments.

**Reviewer Scores:**

Reviewer 1 (Score: 4): Would likely maintain at 4, as discussion would reinforce concerns about weak theoretical foundation and incremental novelty.
Reviewer 2 (Score: 8): Would likely maintain at 8.
Reviewer 3 (Score: 4): Would likely maintain at 4, as their concerns about alignment and data quality bottlenecks are architectural and remain unaddressed.

---

### Decision · Program_Chairs · 2026-01-26

Reject